# Impaired retrograde transport of axonal autophagosomes contributes to autophagic stress in Alzheimer's disease neurons

**Prasad Tammineni[†], Xuan Ye[†], Tuancheng Feng, Daniyal Aikal, Qian Cai\***

Department of Cell Biology and Neuroscience, Rutgers, The State University of New Jersey, Piscataway, United States

**Abstract** Neurons face unique challenges of transporting nascent autophagic vacuoles (AVs) from distal axons toward the soma, where mature lysosomes are mainly located. Autophagy defects have been linked to Alzheimer's disease (AD). However, the mechanisms underlying altered autophagy remain unknown. Here, we demonstrate that defective retrograde transport contributes to autophagic stress in AD axons. Amphisomes predominantly accumulate at axonal terminals of mutant hAPP mice and AD patient brains. Amyloid-$\beta$ (A$\beta$) oligomers associate with AVs in AD axons and interact with dynein motors. This interaction impairs dynein recruitment to amphisomes through competitive interruption of dynein-Snapin motor-adaptor coupling, thus immobilizing them in distal axons. Consistently, deletion of *Snapin* in mice causes AD-like axonal autophagic stress, whereas overexpressing Snapin in hAPP neurons reduces autophagic accumulation at presynaptic terminals by enhancing AV retrograde transport. Altogether, our study provides new mechanistic insight into AD-associated autophagic stress, thus establishing a foundation for ameliorating axonal pathology in AD.

**\*For correspondence:** cai@ biology.rutgers.edu

[†]These authors contributed equally to this work

**Competing interests:** The authors declare that no competing interests exist.

## Introduction

Autophagy is the major cellular degradation pathway for long-lived proteins and organelles (*Nixon, 2013*; *Rubinsztein et al., 2011*; *Schneider and Cuervo, 2014*; *Yue et al., 2009*). Altered autophagy has been linked to several major age-related neurodegenerative diseases, including Alzheimer's disease (AD), that are associated with accumulation of misfolded protein aggregates (*Nixon, 2013*; *Rubinsztein et al., 2011*; *Schneider and Cuervo, 2014*; *Yue et al., 2009*). Ultrastructural analysis revealed that AD brains display a unique autophagic stress phenotype: autophagic vacuoles (AVs) massively accumulate and cluster within large swellings along dystrophic neurites (*Nixon et al., 2005*), a typical amyloid $\beta$ (A$\beta$)-associated phenotype not found in other neurodegenerative diseases (*Benzing et al., 1993*). The mechanisms underlying such autophagic stress in AD neurons are largely unknown.

As highly polarized cells with long axons, neurons face the special challenge of transporting AVs containing engulfed aggregated proteins and damaged organelles generated from distal processes toward the soma where mature acidic lysosomes are mainly located (*Nixon, 2013*; *Sheng, 2014*). In neurons, autophagosomes are continuously formed in distal axons (*Maday and Holzbaur, 2014*; *Maday et al., 2012*). Recent studies established that nascent autophagosomes in distal axons move exclusively in the retrograde direction toward the soma for lysosomal proteolysis (*Cheng et al., 2015a*, *2015b*; *Lee et al., 2011a*; *Maday and Holzbaur, 2016*; *Maday et al., 2012*). Such retrograde transport is initiated by fusion of nascent autophagosomes with late endosomes (LEs) into

**eLife digest** Alzheimer's disease is the result of protein fragments called amyloid-$\beta$ peptides accumulating in the brain and forming clumps. These protein "aggregates" disrupt cellular activities and cause serious problems. To combat this process, healthy cells use a process called autophagy to destroy aggregated proteins. The aggregates are first loaded into structures called autophagosomes that then fuse with cell compartments called lysosomes, which contain enzymes that can break down the proteins.

Brain cells called neurons have an unusual shape with branch-like structures and a long projection called an axon that all form off the main cell body. Autophagosomes predominantly form in the axons and need to move toward the cell body where the lysosomes are found. A motor protein called dynein drives the movement of autophagosomes by interacting with an adaptor protein known as Snapin on the surface of these structures. Autophagosomes tend to accumulate within the neurons of individuals with Alzheimer's disease, but it is not known why. Cai et al. examined the ability of autophagosomes to move to the cell body of neurons from a mouse model of Alzheimer's disease in which human amyloid-$\beta$ peptides accumulate in the brain, and in the brains of human patients with Alzheimer's disease.

The experiments show that autophagosomes predominantly accumulate in the axons and at the ends of axons during Alzheimer's disease. Amyloid-$\beta$ aggregates associate with autophagosomes in the axons and interact with dynein motors. This disrupts the interaction between dynein and Snapin and impairs dynein binding to the autophagosomes, trapping the autophagosomes in the axons. Increasing the production of Snapin proteins inside the mouse neurons enhances dynein binding to autophagosomes and thus helps these structures move to the cell body.

The next step is to investigate whether increasing the ability of autophagosomes to move to the cell body reduces the symptoms of Alzheimer's disease in the mutant mice. This will help to build a foundation for the future development of new strategies to treat Alzheimer's disease and other neurodegenerative disorders that are caused by protein aggregates.

amphisomes and is driven by LE-loaded dynein-Snapin (motor-adaptor) complexes (*Cheng et al., 2015a*). The unique accumulation of immature AVs in the dystrophic neurites of AD brains therefore raises the fundamental question of whether the AV transport and maturation events are affected in AD.

Axonal transport defects have been implicated in AD (*Pigino et al., 2009*, *2003*; *Stokin et al., 2005*; *Tang et al., 2012*). A$\beta$ has been shown to interfere with axonal transport (*Decker et al., 2010*; *Hiruma et al., 2003*; *Pigino et al., 2009*; *Rui et al., 2006*; *Tang et al., 2012*; *Vossel et al., 2010*). Many studies have been focusing on the mechanisms underlying interruption of kinesin-mediated anterograde transport by A$\beta$, raising the fundamental questions as to whether dynein-mediated retrograde transport is also impaired in AD neurons and, if so, whether such transport defects contribute to AD-associated autophagic stress.

Intracellular A$\beta$ accumulation is closely correlated with the progression of AD in the early stages of AD (*LaFerla et al., 2007*; *Li et al., 2007*). A$\beta$ was shown to be generated in the ER and Golgi and also trafficked into the cytosol via the endocytic pathway or passive transport, leading to the accumulation of intracellular A$\beta$ (*Gouras et al., 2005*; *LaFerla et al., 2007*). Intracellular A$\beta$ is enriched in both AD human brains and AD mouse models in association with dystrophic neurites and abnormal synaptic morphology (*LaFerla et al., 2007*; *Spires-Jones and Hyman, 2014*). Intracellular A$\beta$ was proposed to induce presynaptic dysfunction, which might be one of the pathophysiological origins of early AD (*Parodi et al., 2010*; *Yang et al., 2015*). Several lines of evidence indicate that A$\beta$1-42 is associated with LEs or multivesicular bodies (MVBs) and AVs in AD brains (*Takahashi et al., 2004*, *2013*, *2002*; *Yu et al., 2005*). Given that AD-linked autophagic stress is uniquely associated with A$\beta$ generation and amyloid pathology (*Benzing et al., 1993*; *Nixon, 2007*), this raises an important question as to whether intracellular A$\beta$ accumulation augments AV retention in distal AD axons and impairs autophagic clearance.

In the current study, we provide new evidence that AVs aberrantly accumulate in distal axons and at the presynaptic terminals of mutant hAPP Tg mice and AD patient brains. Amphisomes, rather than autophagosomes are predominantly retained within axons. We demonstrate impaired retrograde transport of amphisomes in live AD axons. Soluble A$\beta$42 oligomers are enriched in the distal axons of AD mouse brains accompanied by the accumulation of amphisomes. Moreover, we reveal for the first time the molecular interruption leading to such transport defects in AD neurons: direct interaction of oligomeric A$\beta$1-42 with dynein intermediate chain (DIC) disrupts the coupling of dynein-Snapin, a motor-adaptor complex essential for recruiting dynein transport machinery to LEs and amphisomes. This mechanism is further confirmed in *Snapin* KO mice. *Snapin* deficiency impedes the removal of AVs from distal axons and synapses and recapitulates AD-associated autophagic stress. More importantly, overexpression Snapin in mutant hAPP Tg neurons reduces autophagic retention in distal axons and presynaptic terminals by enhancing their retrograde transport. Snapin mutant defective in DIC-binding fails to rescue autophagic stress in AD axons, thus supporting our conclusion that defective retrograde transport is one of main mechanisms underlying the AD-linked autophagic stress. Thus, our study provides new mechanistic insights into how A$\beta$ impairs dynein-mediated retrograde transport of LEs and amphisomes, thus leading to autophagic pathology in AD axons. Our study also establishes a foundation for future investigation into regulation of dynein-Snapin coupling to attenuate autophagic defects in AD brains.

## Results

### Autophagic accumulation in the distal axons of mutant hAPP Tg mouse brains

To determine whether autophagy is altered in AD neurons, we first examined the hippocampi of both wild-type (WT) and hAPP transgenic (Tg) mice harboring the human AD Swedish and Indiana mutations (*Camk2α*-tTA X tet-APPswe/ind) (*Jankowsky et al., 2005*). In WT mouse brains, the autophagic marker LC3 appeared as a diffused pattern in the hippocampal mossy fiber processes. However, in mutant hAPP Tg mouse brains, a majority of LC3 associated with vesicular structures reflecting clustered autophagic vacuoles (AVs). The average number of LC3-labeled AV clusters per slice section was substantially increased relative to that of WT mouse brains (WT: $7.0 \pm 1.028$; mutant hAPP Tg: $53.35 \pm 3.78$; $p < 1 \times 10^{-10}$) (*Figure 1A,B*). Given that the hippocampal mossy fibers are composed of axons and presynaptic terminals from granule cells in the dentate gyrus, this observation suggests aberrant accumulation of AVs in the axons and axonal terminals of mutant hAPP Tg mouse brains. To further test this possibility, we performed additional line of experiments and showed that AVs accumulated in the distal axons surrounding amyloid plaques (*Figure 1—figure supplement 1A,B*). While 21.05% of LC3-marked AVs localized to MAP2-labeled dendrites, 83.11% and 91.64% of AVs co-localized with the presynaptic marker synaptophysin and along Neurofilament (NF)-labeled axons, suggesting that autophagic stress occurs predominantly in the axons and presynaptic terminals of AD mouse brains (*Figure 1—figure supplement 1A,B*).

To determine those AVs as autophagosomes or amphisomes following fusion with late endocytic organelles, we next performed co-immunstaining with an antibody against cation-independent mannose 6-phosphate receptor (CI-MPR), a membrane protein preferentially located in late endosomes (LEs) (*Griffiths et al., 1988*). Consistent with previous studies (*Takahashi et al., 2004*, *2002*; *Ye and Cai, 2014*), CI-MPR-labeled late endocytic organelles abnormally accumulated along the neuronal processes of AD mouse brains (*Figure 1D*). Surprisingly, the majority of LC3-labeled AVs co-localized with LEs in the hippocampal mossy fibers of AD mouse brains (*Figure 1D*), suggesting those AVs as amphisomes in nature following fusion with LEs. The average number of amphisomes labeled by both LC3 and CI-MPR per slice section was significantly increased compared to WT mouse brains (WT: $7.28 \pm 0.62$; mutant hAPP Tg: $45.41 \pm 2.75$; $p < 1 \times 10^{-12}$) (*Figure 1C*). Moreover, a significant number of LC3 clusters co-labeled with Ubiquitin, p62, or CI-MPR were retained in the hippocampal mossy fibers and within swollen/dystrophic axons surrounding amyloid plaques (Ubiquitin: $97.89\% \pm 0.23\%$; p62: $96.29\% \pm 0.43\%$; CI-MPR: $93.68\% \pm 0.51\%$) (*Figure 1—figure supplement 1C–E*). Our data suggests predominant accumulation of amphisomes in the distal axons of AD mouse brains.

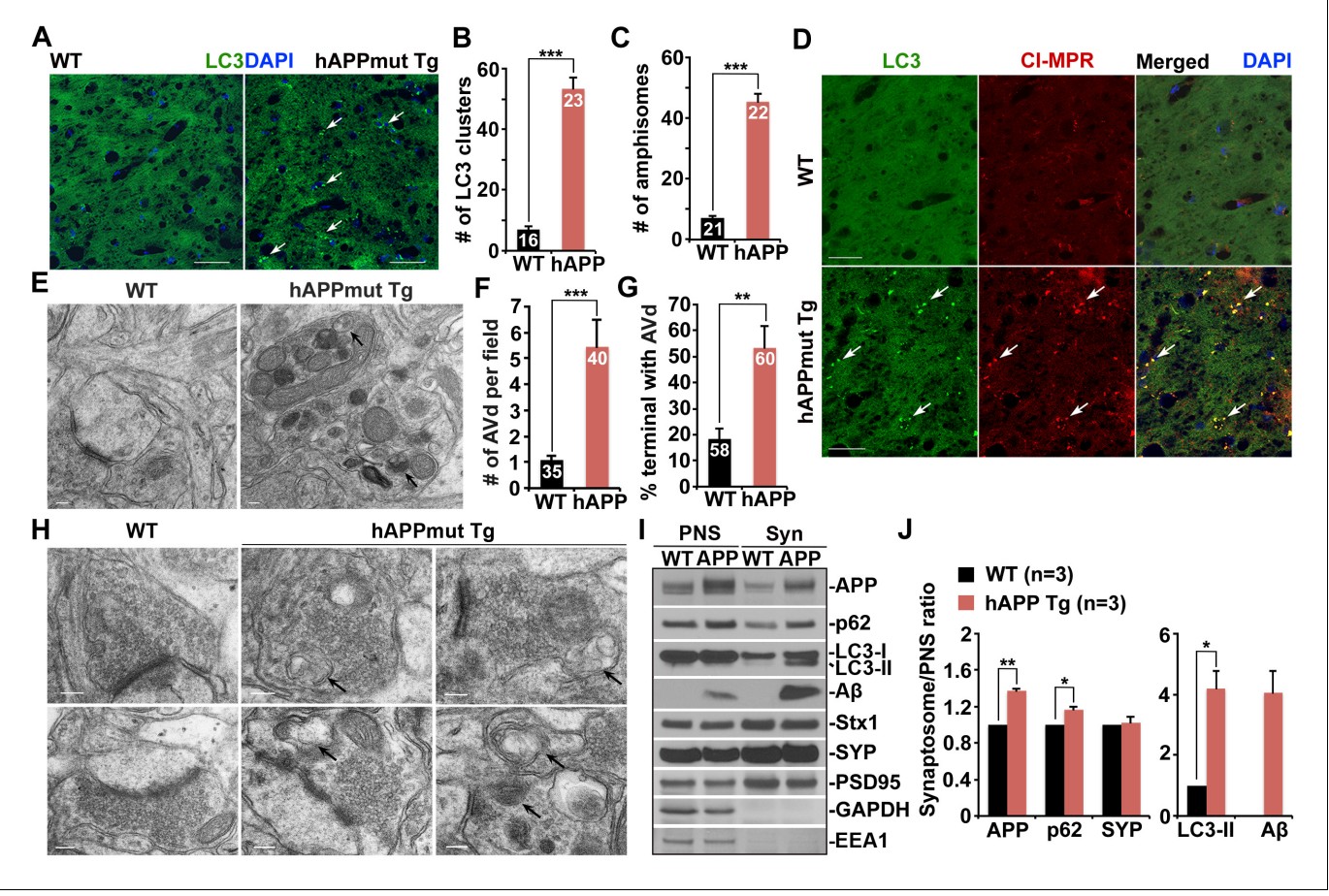

**Figure 1.** Autophagic accumulation in the distal axons of mutant hAPP Tg mice. (**A** and **B**) Representative images (**A**) and quantitative analysis (**B**) showing accumulation of LC3-labeled autophagic vacuoles (AVs) in the hippocampal mossy fibers of eight-month mutant hAPP Tg mice. (**C** and **D**) Quantitative analysis (**C**) and representative images (**D**) showing amphisome retention in the hippocampal mossy fibers of hAPP mice. Note that LC3-labeled AV clusters were co-localized with cation-independent mannose 6-phosphate receptor (CI-MPR), a late endosome (LE) marker, suggesting that those AVs are amphisomes in nature following fusion with LEs. (**E** and **F**) Representative TEM images (**E**) and quantitative analysis (**F**) showing abnormal retention of AVd-like organelles within enlarged neurites in the hippocampal regions of mutant hAPP Tg mouse brains. Note that dystrophic/swollen neurites contained predominantly AVd-like structures marked by arrows, which was not readily observed in wild-type (WT) mice. The average number of AVd per EM field was quantified. (**G** and **H**) Quantitative analysis (**G**) and representative TEM images (**H**) showing aberrant accumulation of AVd-like structures (black arrows) at presynaptic terminals in hAPP mice. AVd-like structures, indicated by arrows, were not readily observed in WT mouse brains. Percentage of presynaptic terminals containing AVd was quantified. (**I** and **J**) Abnormal synaptic retention of LC3-II and p62 (autophagy markers), APP, and Aβ in mutant hAPP Tg mouse brains. Equal amounts (15 μg) of synapse-enriched synaptosomal preparations (Syn) and post-nuclear supernatants (PNS) from WT and hAPP mice were sequentially immunoblotted on the same membrane after stripping between each antibody application. The purity of synaptosomal fractions was confirmed by the absence of EEA1 and GAPDH. The synaptosome/PNS ratio in AD mice was compared to those in WT littermates. Data were quantified from three independent repeats. Stx1: syntaxin 1; SYP: synaptophysin Scale bars: 25 μm (**A** and **D**), 100 nm (**E**), and 200 nm (**H**). Data were quantified from a total number of imaging slice sections indicated on the top of bars (**B** and **C**) from three pairs of mice. The average numbers of AV clusters per section (320 μm × 320 μm) and per EM field (10 μm × 10 μm) were quantified (**B** and **C**). Error bars represent SEM. Student's *t* test: ***p<0.001, **p<0.01, *p<0.05.

The following figure supplement is available for figure 1:

**Figure supplement 1.** Axonal autophagic stress in mutant hAPP Tg mouse brains.

Using Transmission Electron Microscopy (TEM), we examined AVs at the ultrastructual level based on the established AV morphological features: initial AVs (AVi) contain intact cytosol and/or organelles with a sealed double-membrane bilayer separated by an electron-lucent cleft, whereas late-stage degradative AVs (AVd) after fusion with late endocytic organelles are those containing small

internal vesicles and/or organelles at various stages of degradation, electron-dense amorphous material (*Cheng et al., 2015a*; *Klionsky et al., 2012*). The AV-like structures were observed within dystrophic/swollen neurites in mutant hAPP mouse brains: most of those were AVd-like structures that were not readily found in WT mouse brains (*Figure 1E*). We examined the frequency and average number of AVd-like organelles per EM field within the hippocampal regions of AD mice. EM images containing cell bodies were excluded from analysis. Over 70% of images from WT mice have zero or one AV present, whereas more than 90% of EM images from hAPP Tg mice have at least one AV per field (*Figure 1—figure supplement 1F*). Moreover, compared to WT (1.09 ± 0.18; n = 35), AD mice also displayed an increased incidence of AVs per EM field (5.45 ± 1.05; n = 40; p=0.000632) (*Figure 1F*). This observation is consistent with our immunostaining results (*Figure 1A–D*), suggesting autophagic accumulation in AD mouse brains.

We next assessed AV distribution in the axonal terminals of AD mice. We found a striking number of AVd-like structures were retained within the presynaptic terminals of AD mice (53.33% ± 8.43%; n = 60; p=0.0069) relative to that of WT controls (18.33% ± 4.01%; n = 58) (*Figure 1G,H*). To confirm these ultrastructural observations, we purified synapse-enriched synaptosomes using Percoll gradient centrifugation as previously described (*DiGiovanni et al., 2012*). AD mouse brains displayed significantly increases of the synaptosomal preparations (Syn)/post-nuclear supernatants (PNS) ratio in APP (1.36 ± 0.03; p=0.007319), p62 (1.17 ± 0.02; p=0.021517), and LC3-II (4.2 ± 0.57; p=0.011328), but not SYP (1.02 ± 0.07; p=0.73211) relative to those of WT littermates. The levels of human $A\beta$ detected by 6E10 antibody showed a four-fold increase in synaptosomal fractions compared to that of PNS fractions in AD mice (*Figure 1I,J*). This result suggested that APP, p62, LC3-II, and $A\beta$ are relatively enriched in the synaptic terminals of AD mice. Altogether, our TEM and light imaging data combined with biochemical analysis consistently indicate that amphisomes predominantly accumulate in the distal axons of AD mouse brains.

## Impaired retrograde transport of axonal amphisomes in mutant hAPP Tg neurons

Many recent studies demonstrated that autophagosomes are predominantly generated in distal axons, and undergo exclusively retrograde transport following fusion with LEs to form amphisomes for lysosomal proteolysis in the soma (*Cheng et al., 2015a*; *Wong and Holzbaur, 2015*). We next assessed the distribution of AVs in cultured live cortical neurons from WT and mutant hAPP Tg mice harboring the human AD Swedish and Indiana mutations (J20) (*Mucke et al., 2000*). Neurons were co-transfected with autophagy marker GFP-LC3 and LE marker mRFP-Rab7, followed by imaging at DIV17-19. In WT neurons, GFP-LC3 was diffused in the cytoplasm in the form of cytosolic LC3-I, whereas Rab7-labeled LEs appeared as vesicular structures along axonal processes (*Figure 2A*). To our surprise, GFP-LC3 associated with vesicles as lipidated LC3-II in mutant hAPP Tg neurons at basal condition (*Figure 2B*). Similar to WT neurons, the majority of autophagosomes in AD neurons co-localized with Rab7-labeled LEs along the axons (WT: 80.64% ± 4.36%; hAPP: 86.12% ± 2.48%; p=0.27849) (*Figure 2—figure supplement 1A*), suggesting effective formation of amphisomes by fusion of these two organelles. However, compared to WT neurons, the density of axonal AVs, particularly amphisomes, was robustly increased in AD neurons (autophagosomes per 100 µm length: WT 0.47 ± 0.11; hAPP 1.52 ± 0.39; p=0.01306; amphisomes per 100 µm length: WT 2.17 ± 0.23; hAPP 8.97 ± 0.68; p<$1\times10^{-12}$) (*Figure 2—figure supplement 1A*). Axonal AV accumulation in hAPP neurons was further confirmed by its negative staining for MAP2 (*Figure 2—figure supplement 1B*).

We next examined cortical neuron ultrastructure at DIV18-19 from WT and mutant hAPP Tg mice. A striking number of AV-like organelles were found along AD neurites (1.68 ± 0.10 per 10 µm length; p<$1\times10^{-16}$), a phenotype rarely detected in WT neurons (*Figure 2C,D*). Consistently, we demonstrated an increased percentage of presynaptic terminals in AD neurons containing AVs (35.20% ± 5.0%; n = 58 EM fields; p<$1\times10^{-6}$) relative to that of WT controls (8.98% ± 2.41%; n = 54) (*Figure 2C,E*). Thus, these axonal imaging data from cultured neurons are consistent with our in vivo evidence from AD mouse brains (*Figure 1* and *Figure 1—figure supplement 1*), suggesting axonal accumulation of amphisomes. We also showed that the majority of those axonal AVs were co-labeled with mRFP-Ubiquitin (Ub) (hAPP: 91.67% ± 1.26% from 49 axons; WT: 87.5% ± 7.14% from 20 axons; p=0.6331) (*Figure 2—figure supplement 1E–G*). Thus, consistent with our observations in AD mouse brains (*Figure 1—figure supplement 1C–E*), this result indicates that these AVs contained engulfed ubiquitinated cargoes.

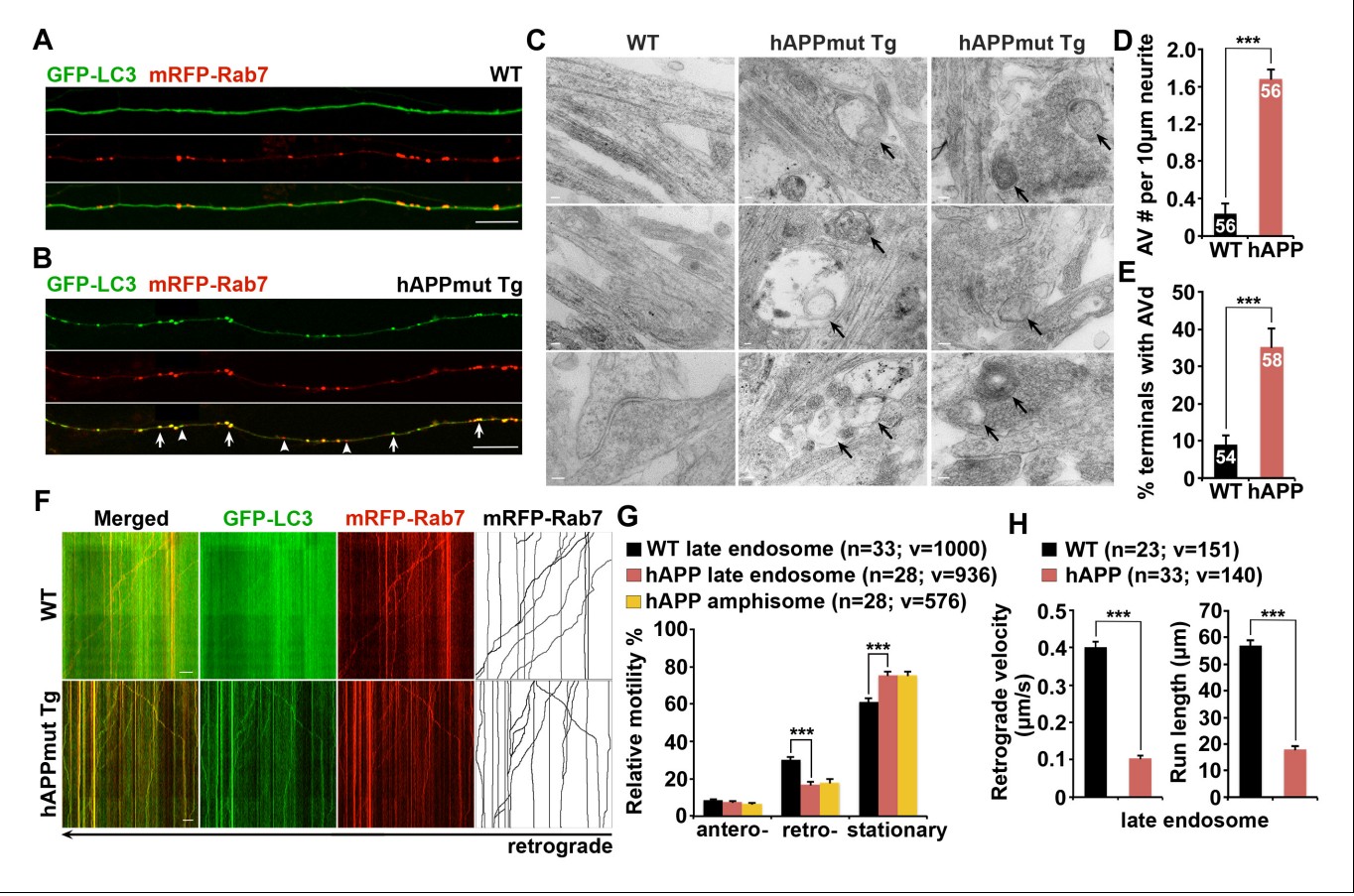

**Figure 2.** Impaired retrograde transport of axonal amphisomes in mutant hAPP Tg neurons. (A and B) Axonal amphisomes predominantly accumulated in cultured cortical neurons derived from mutant hAPP Tg mice. Cortical neurons were co-transfected with GFP-LC3 and mRFP-Rab7, followed by imaging at DIV17-19. Images were taken from the distal axons of WT (A) and mutant hAPP Tg neurons (B). Late endosomes (LEs) are positive for Rab7 alone, whereas amphisomes are positive for both Rab7 and LC3. Arrow indicates amphisome co-labeled with LC3 and Rab7. Arrowhead points to AV or LE alone. (C–E) Representative TEM images (C) and quantitative analysis (D and E) showing aberrant accumulation of AVs in neuronal processes and presynaptic terminals of mutant hAPP Tg neurons. TEM showing retention of AVd-like organelles at the axonal terminals of hAPP neurons at DIV18-19. Arrows indicate AVd-like structures, which were not readily observed in WT neurons. Images were representative from 50–150 electron micrographs of neurons cultured from three pairs of WT and mutant hAPP mice. (F–H) Dual-channel kymographs showing impaired retrograde transport of amphisomes in hAPP neurons. Vertical lines represent stationary organelles. Slanted lines or curves to the right (negative slope) represent anterograde movement; to the left (positive slope) indicate retrograde movement. An organelle was considered stationary if it remained immotile (displacement ≤5 μm). GFP-LC3 was diffused and LEs predominantly moved toward the soma in WT neurons, whereas the majority of amphisomes (labeled by both LC3 and Rab7) remained stationary in the axons of hAPP neurons (F). Note that amphisomes and LEs share similarly reduced retrograde motility in the same axons of hAPP neurons. Relative motility of LC3-labeled AVs in hAPP neurons and LEs in WT neurons and hAPP neurons were examined (G). The average velocity and run length of LE retrograde transport in WT and hAPP neurons were quantified (H). Data were quantified from the total number of vesicles (v) in the total number of neurons (n) indicated in parentheses from more than four experiments. Scale bars: 100 nm (C), 5 μm (A and B), and 10 μm (F), and. Error bars represent SEM. Student's *t* test: ***p<0.001, **p<0.01, *p<0.05.

The following figure supplement is available for figure 2:

**Figure supplement 1.** Axonal accumulation of amphisomes containing engulfed ubiquitinated cargoes in mutant hAPP Tg neurons.

The aberrant AV retention in the distal AD axons may reflect defects in their retrograde transport toward the soma, thus reducing autophagic clearance. We next assessed the retrograde motility of axonal AVs in live mutant hAPP Tg neurons. In WT neurons, while GFP-LC3 was diffuse, a significant portion of Rab7-labled LEs (30.16% ± 1.33%) moved in the retrograde direction toward the soma along the same axon (*Figure 2F,G*). However, LEs in AD neurons displayed reduced retrograde motility in distal axons (16.86% ± 1.51%; p<1×10$^{-8}$). Such reduction was not found in anterograde

transport of LEs (p=0.35506) (*Figure 2G*). Strikingly, amphisomes displayed the similar motility pattern: reduced retrograde (18.04% ± 1.74%), but not anterograde transport in the same axons of AD neurons (*Figure 2F,G*). We also quantified the average retrograde velocity and run length of Rab7-marked organelles in WT and hAPP mutant neurons (*Figure 2H*). Consistent with previous studies (*Castle et al., 2014*; *Deinhardt et al., 2006*), the average retrograde velocity and run length of Rab7-associated LEs in WT neurons were 0.40 ± 0.01 μm/sec and 56.97 ± 2.02 μm, respectively, which were significantly reduced in AD neurons (velocity: 0.10 ± 0.007 μm/sec, $p<1\times10^{-14}$; run length: 17.95 ± 1.18 μm, $p<1\times10^{-12}$). Altogether, these observations indicate that aberrant accumulation of amphisomes in distal AD axons may result from impaired retrograde transport. Moreover, we showed increased levels of APP (2.05 ± 0.29; p=0.015952) and C99 (6.13 ± 1.07; p=0.008705), but not C83 (1.18 ± 0.10; p=0.16055) in mutant hAPP Tg neurons relative to those of neurons from WT littermates (*Figure 2—figure supplement 1C,D*).

We also examined the co-localization of LC3 with Rab5-labeled early endosomes in cultured neurons from mutant hAPP Tg mice. We found that about 46% of LC3-labeled AVs co-localized with early endosomes within the axon of mutant hAPP neurons (45.58% ± 2.24%; n = 47, v = 875) (*Figure 2—figure supplement 1H*). However, Rab5-marked early endosomes moved either a short distance, or in an oscillatory pattern along axons (*Figure 2—figure supplement 1I*). While our observation is consistent with the results from previous studies (*Cai et al., 2010*; *Chen and Sheng, 2013*), the motility of axonal early endosomes showed no significant change in mutant hAPP neurons relative to that of WT neurons (WT: 67.53% ± 1.97; hAPP: 70.93% ± 2.31%; p=0.268) (*Figure 2—figure supplement 1J*). A recent study reported that nascent AVs gain retrograde transport motility by recruiting LE-loaded dynein-Snapin motor-adaptor complexes after fusion with Rab7-associated LEs to form amphisomes (*Cheng et al., 2015a*, *2015b*). Thus, our data supports the notion that fusion of AVs with Rab5-endosomes could be a transitional process before they further mature into Rab7-positive amphisomes to gain long-distance retrograde transport motility.

## Association of soluble Aβ oligomers with amphisomes in the dystrophic axons of AD mice

Given that AD-associated autophagic stress is uniquely linked to Aβ generation and amyloid pathology (*Benzing et al., 1993*; *Nixon, 2007*), we next addressed whether Aβ associates with these AVs in the distal axons of AD neurons. We showed that the majority of LC3-labeled AVs co-localized with anti-β amyloid antibody (6E10)-labeled APP and Aβ within dystrophic neurites, but not with the core of 6E10 antibody-marked fibrillar amyloid plaques (*Figure 3A*). The co-localization of LC3-labeled AVs with 6E10 within neurites is 74.9% ± 2.37% in the hippocampal regions of AD mice. It is predictable that not all AVs are associated with APP and Aβ and that some autophagosomes may function in engulfing other autophagic cargos such as dysfunctional organelles and cytosolic components.

Next, to examine the association of Aβ oligomers with AVs in AD brains, we utilized the well-characterized A11 antibody that only recognizes soluble Aβ oligomers, but not APP and its cleaved products C99, soluble monomer, or insoluble fibrils (*Jimenez et al., 2008*, *2011*; *Kayed et al., 2003*; *Zempel et al., 2010*). We found that the A11 antibody preferentially labeled soluble Aβ oligomers within neurites at the periphery of amyloid plaques indicated by 6E10 (*Figure 3B*). This observation suggests that A11-marked soluble Aβ oligomers surrounds fibrillar plaque cores (*Spires-Jones and Hyman, 2014*). Oligomeric Aβ is more enriched in axons and associated with amphisomes within the dystrophic axons of AD mice (*Figure 3B–E*). The percentage of A11-labeled oligomeric Aβ co-localization is 91.0% ± 1.21% with 6E10, 89.5% ± 3.84% with NF, 88.0% ± 1.33% with CI-MPR, 90.0% ± 1.18% with p62, and 89.0% ± 1.16% with Ubiquitin, respectively. Amphisomes are also concentrated in NF-labeled distal axons (the percentage of co-localization with NF: 82.61% ± 0.86% (CI-MPR); 80.74% ± 0.72% (Ubiquitin)) (*Figure 3—figure supplement 1C,D*). Using another well-characterized antibody (AB5078P) recognizing soluble Aβ1-42 oligomers, but not Aβ1-40 or high molecular weight insoluble forms of Aβ1-42 (*Agholme et al., 2012*; *Kamal et al., 2001*; *Muresan et al., 2009*; *Takahashi et al., 2013*), we observed similar results: the co-localization of oligomeric Aβ1-42 is 92.5% ± 1.66% with 6E10, 90.4% ± 1.21% with CI-MPR, 88.9% ± 1.79% with p62, and 93.2% ± 1.26% with Ubiquitin, respectively (*Figure 3—figure supplement 1A,B*). 91.86% ± 0.38% of Aβ1-42 co-localized with NF-labeled distal axons surrounding amyloid plaques (*Figure 3—figure supplement 1A,B*).

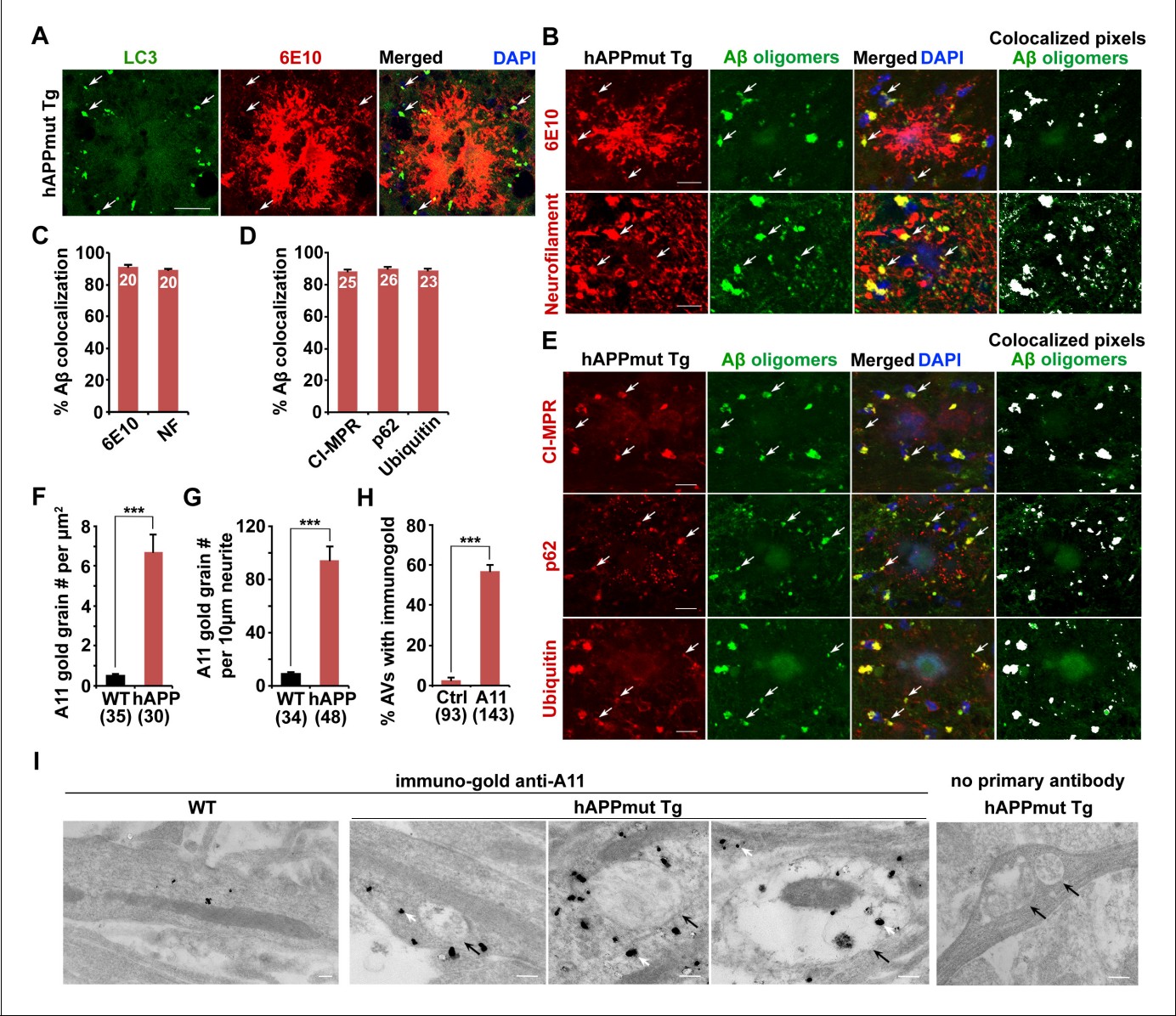

**Figure 3.** Association of soluble Aβ oligomers with amphisomes in the dystrophic axons of AD mice. (**A**) AVs clustering within swollen/dystrophic neurites around an amyloid plaque enriched with 6E10 antibody-labeled APP, C99, or Aβ deposits. (**B** and **C**) Representative images (**B**) and quantitative analysis (**C**) showing that soluble Aβ oligomers labeled by anti-A11 antibody was concentrated within Neurofilament (NF)-labeled axons surrounding amyloid plaques in mutant hAPP Tg mice. The percentage of soluble Aβ co-localization with 6E10 antibody-labeled Aβ or NF was quantified, respectively. (**D** and **E**) Quantitative analysis (**D**) and representative images (**E**) showing the association of soluble Aβ oligomers with amphisomes within dystrophic axons around amyloid plaques in the hippocampal regions of mutant hAPP mice. The percentage of oligomeric Aβ co-localization with CI-MPR, p62, and Ubiquitin (Ub) was quantified, respectively. (**F–I**) Immuno-EM analysis (**I**) and quantification (**F**, **G**, and **H**) showing that soluble Aβ oligomers in the cytoplasm, marked by anti-A11 immuno-gold particles (white arrows), associated with or surround AVd-like structures (black arrows) within the neurites of cultured mutant hAPP Tg neurons. Note that anti-A11 immuno-gold particles were also present within the AVd-like structures containing organelles and small vesicles along hAPP neurites. Anti-A11 immuno-gold particles were detected in the cytoplasm of neurites in WT neurons. The average numbers of the A11 gold grains per EM field (10 μm × 10 μm) and per 10 μm neurite were quantified in WT and hAPP neurons, respectively. The percentage of AVd-like compartments surrounded by the gold particles was quantified in hAPP neurons or in the absence of the primary antibody. The co-localized pixels of individual markers with Aβ oligomers were indicated in white (**B** and **E**). Data were quantified from a total number of imaging slice sections (320 μm × 320 μm) indicated on the top of bars (**C** and **D**) and from a total number of EM fields (**F**), neurites (**G**), or AVs (**H**) indicated in parentheses from more than three experiments. Scale bars: 25 μm (**A**), 10 μm (**B** and **E**), and 200 nm (**I**). Error bars represent SEM. Student's *t* test: ***p<0.001, **p<0.01, *p<0.05.

The following figure supplement is available for figure 3:

*Figure 3 continued on next page*

Figure 3 continued

**Figure supplement 1.** Association of soluble Aβ1-42 oligomers with amphisomes in the distal axons of mutant hAPP Tg mice.

To further confirm these imaging results at the ultrastructural level, we performed immuno-EM analysis in cultured WT and mutant hAPP Tg neurons using the A11 antibody that detect soluble Aβ oligomers. Consistent with a previous study (*Diomede et al., 2014*), we found that the immuno-gold anti-A11 antibody-labeled oligomeric Aβ was mostly present in the cytoplasm of WT, a pattern similarly found in hAPP neurons (*Figure 3I*). However, the average numbers of anti-A11 immuno-gold grains per μm² EM field and per 10 μm neurites were markedly increased in AD neurons relative to those of WT littermate controls (EM field: WT: 0.53 ± 0.06; hAPP: 6.71 ± 0.86; p<1×10⁻⁸; Neurite with gold grains: WT: 9.19 ± 0.91; hAPP: 94.54 ± 10.07; p<1×10⁻¹⁰) (*Figure 3F,G*), suggesting that Aβ oligomers is enriched in the cytoplasm of AD axons. Moreover, ~57.3% of AVd-like structures were associated with or surrounded by oligomeric Aβ gold particles in the cytoplasm of hAPP axons (*Figure 3H*). We also detected the presence of anti-A11 immuno-gold within AVd-like structures. Altogether, our data consistently indicate amphisome-associated Aβ1-42 oligomers in distal axons of AD neurons.

## Oligomeric Aβ42-Mediated interruption of Dynein-Snapin coupling and recruitment of dynein motors to amphisomes

Dynein is the primary motor that drives retrograde transport of both LEs and AVs from distal axons to the soma (*Cai et al., 2010*; *Lee et al., 2011a*; *Maday et al., 2012*). Our previous study revealed that Snapin serves as an adaptor for the recruitment of dynein motors to LEs by binding to DIC (*Cai et al., 2010*). Disrupting DIC-Snapin coupling impairs LE retrograde transport. We recently established that autophagosomes acquire their retrograde motility by recruiting LE-loaded dynein-Snapin (motor-adaptor complex) upon fusion of these two organelles (*Cheng et al., 2015a*). We next asked whether reduced AV retrograde transport is caused by impaired dynein motor recruitment. By immunoisolation of LEs/amphisomes using anti-Rab7-coated magnetic beads, we detected reduced attachment of dynein DIC onto the purified LEs/amphisomes in mutant hAPP mouse brains relative to WT littermates (0.27 ± 0.02; p=0.000686) (*Figure 4A,B*). Conversely, there was a significant increase in LC3-II levels (p=0.01061) along with enhanced hAPP in the purified Rab7-associated organelles of mutant hAPP mouse neurons (*Figure 4A,B*), suggesting aberrant amphisomal retention. As controls, similar amounts of Rab7 and Snapin were associated with LEs from WT and mutant hAPP Tg mouse brains. Our study suggests that reduced attachment of dynein motors impairs AV retrograde transport of amphisomes toward the soma, thus leading to AV retention in distal AD axons.

We next examined whether Aβ overproduction interrupts dynein-Snapin coupling, and thus impedes AV transport in AD neurons. By co-immunoprecipitation assays we observed that less dynein DIC-Snapin complexes were detected in cells expressing mutant hAPP^swe relative to WT hAPP (*Figure 4C*), which suggests that Aβ overproduction interferes with the assembly of the DIC-Snapin complex. Next, we sought to ask whether Aβ interacts with dynein DIC or Snapin, thus competitively interrupting the dynein-Snapin coupling. To address this question, we prepared soluble Aβ1-42 oligomers and performed five lines of experiments. First, direct in vitro binding of Aβ1-42 to GST-DIC, but not GST-Snapin or GST, was detected (*Figure 4D*). GST-DIC specifically interacts with Aβ1-42, but not Aβ1-40 or scrambled Aβ (*Figure 4E,F*). Second, Aβ1-42 interaction with DIC was increased as Aβ concentration was elevated from 0.5 μM to 4 μM in the presence of the same amount of GST-DIC (*Figure 4G*). Third, as low as 0.2 μM Aβ1-42 is effective in interference with the assembly of DIC-Snapin complex; this interruption is in a dose-dependent manner when the same amount of Snapin and DIC was used (*Figure 4H*). Fourth, DIC bound to oligomeric Aβ1-42 (*Figure 4I*), the toxic form of Aβ linked to AD pathogenesis. Fifth, oligomeric Aβ1-42 interfere with DIC interaction with LE/amphisome-associated Snapin, thus impairing the recruitment of dynein motors to LEs and amphisomes prepared from mouse brains (*Figure 4J*). Sixth, we demonstrated a DIC-Aβ complex in mutant hAPP Tg mouse brains by immunoprecipitation (*Figure 4K*). Thus, our findings suggest that dynein-Snapin coupling, and thus cargo-motor association, is compromised by

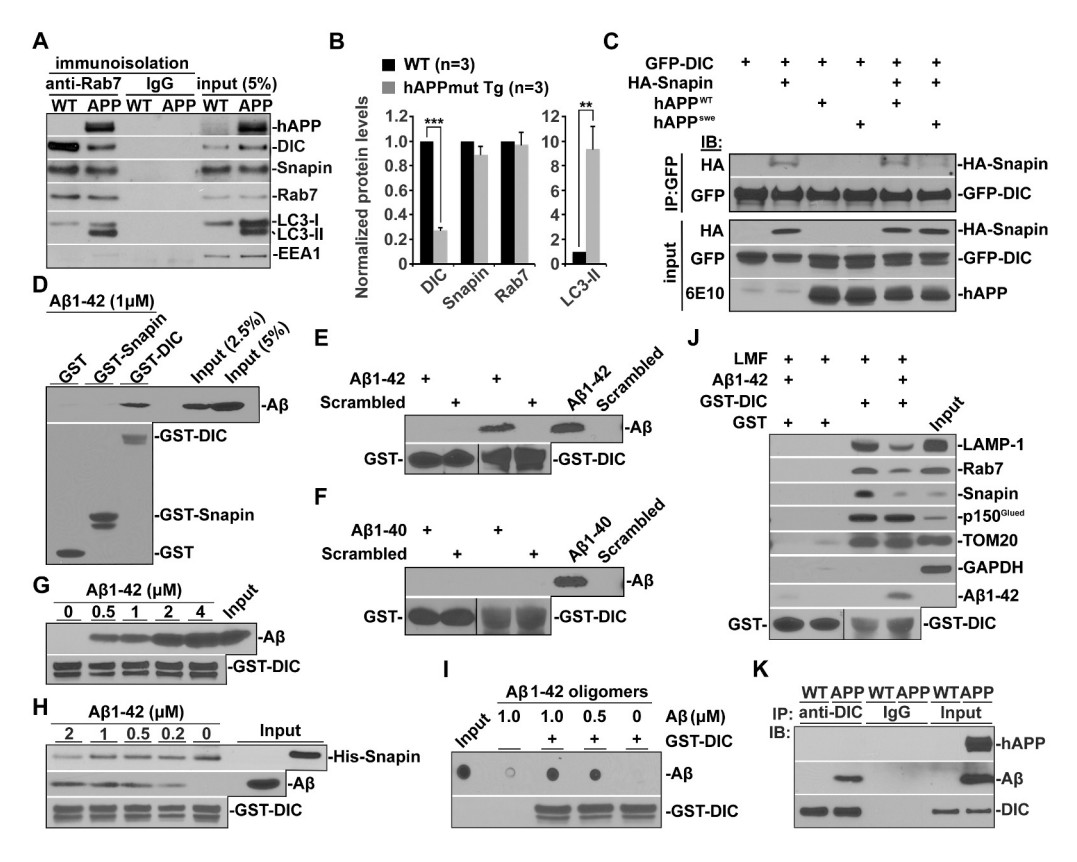

**Figure 4.** Oligomeric Aβ42-mediated interruption of dynein-Snapin coupling and recruitment of dynein motors to amphisomes. (A and B) Immunoisolation assays (A) and quantitative analysis (B) from three repeats showing reduced dynein attachment to amphisomes in sixteen-month mutant hAPP Tg mouse brains. Rab7-associated organelles were immunoisolated from light membrane fractions, followed by sequential immunoblotting on the same membranes with antibodies against the dynein intermediate chain (DIC), LC3, hAPP, Snapin, Rab7, and EEA1. Note that AD mouse brains exhibited reduced DIC and increased LC3-II levels in the purified amphisomal organelles. Data were quantified from three independent repeats of three pairs of mice. (C) Immunoprecipitation showing reduced Snapin-DIC coupling in COS7 cells expressing mutant hAPP^swe, but not WT hAPP. (D) Direct interaction of Aβ1-42 with GST-DIC, but not GST-Snapin or GST. 6E10 antibody was used to detect Aβ. (E and F) GST-DIC specifically interacts with Aβ1-42, but not Aβ1-40 or scrambled Aβ. (G) Aβ1-42 interaction with DIC was increased as Aβ concentration was elevated from 0.5 μM to 4 μM in the presence of the same amount of GST-DIC. (H) Aβ1-42 interferes with DIC-Snapin coupling in a dose-dependent manner. Note that the DIC-Snapin interaction was competitively interrupted in the presence of as low as 0.2 μM Aβ1-42 when the same amount of Snapin and DIC was used. (I) The dot blot using anti-β amyloid antibody showed that GST-DIC bound to oligomeric Aβ. (J) Oligomeric Aβ1-42 interrupts dynein-Snapin coupling and the recruitment of dynein motors to LEs and amphisomes in mouse brains. Membranous organelles were pulled down from light membrane fractions (LMF) of mouse brains by GST-DIC in the presence or absence of 2 μM oligomeric Aβ1-42. Bead-bound membrane organelles were resolved by PAGE and sequentially detected with antibodies on the same membranes after stripping between applications of each antibody. Note that reduced tethering of dynein motors to LAMP-1 or Rab7-associated LEs and amphisomes in the presence of Aβ was specifically caused by impaired DIC-Snapin interaction because the dynein-dynactin (p150^Glued) complex was not affected. The attachment of dynein motors to mitochondria showed no detectable change. TOM20: a mitochondrial outer membrane protein. The purity of the preparation pulled down by GST-DIC beads was confirmed by the absence of GAPDH. Results were representative from three independent repeats. (K) The DIC-Aβ complex was immunoprecipitated by anti-DIC antibody from mutant hAPP Tg mouse brains. Error bars represent SEM. Student's t test: ***p<0.001, **p<0.01, *p<0.05.

accumulation of cytoplasmic Aβ1-42 oligomers, which could be a pathogenic mechanism of impaired dynein-driven axonal transport in AD.

## Impaired Dynein-Snapin coupling contributes to axonal autophagic stress in AD patient brains

AD patient brains display unique autophagic stress characterized by massive AV accumulation within large swellings along dystrophic neurites (*Nixon et al., 2005*), so we next examined whether AVs

abnormally accumulate at nerve terminals in AD patient brains. Strikingly, LC3-II levels were robustly increased in synapse-enriched synaptosomal preparations from AD patient brains relative to those of control subjects (8.05 ± 1.51; p=0.009418) (*Figure 5A,B*). The results were quantified from the experiments using four control subjects and four patient brains (postmortem interval 7.08 hr – 22.5 hr) (*Table 1*). Consistently, aberrant clustering of AVd-like structures was detected within enlarged dystrophic neurites in AD patient brains (*Figure 5C,D*), a phenotype not readily found in control subjects. We quantified the average number of AVd-like organelles per EM field in the brains of two age-matched controls and three AD patients at different Braak stages with postmortem interval between 7.08 hr and 12.5 hr (*Table 1*). Compared to control subjects (1.54 ± 0.16; n = 76 EM fields), AD brains exhibited an increased number of AVd-like structures (8.67 ± 0.79; n = 91; $p<1\times10^{-14}$) (*Figure 5D*). Moreover, immunoprecipitation analysis showed a marked reduction of dynein-Snapin complexes in AD patient brains ($p<1\times10^{-6}$) (*Figure 5E,F*) (*Table 1*). These in vivo observations in AD patient brains further confirm that the interruption of the dynein-Snapin coupling impairs the removal of AVs from distal neurites and synaptic terminals, thus reducing AV clearance by lysosomes in the soma.

## *Snapin*-deficient mouse brains recapitulate AD-associated autophagic stress in axons

Given the observations of dynein-Snapin coupling deficits in AD patient brains, we next asked whether deleting *Snapin* in mice displays autophagic phenotypes similar to those of AD brains. To address this issue, we performed four lines of experiments using *Snapin* flox/flox conditional knock-out (cKO) mice, in which the *Snapin* gene was deleted in the frontal cortex and hippocampus by Cre expression (*Cheng et al., 2015a*; *Ye and Cai, 2014*). First, we examined the distribution pattern of CI-MPR-labeled LEs in the hippocampal CA3 regions. Deletion of *Snapin* leads to LE clustering in the hippocampal mossy fibers composed of axons and presynaptic terminals from granule cells in the dentate gyrus (*Figure 6A*). The majority of these LE clusters were not distributed in the MAP2-

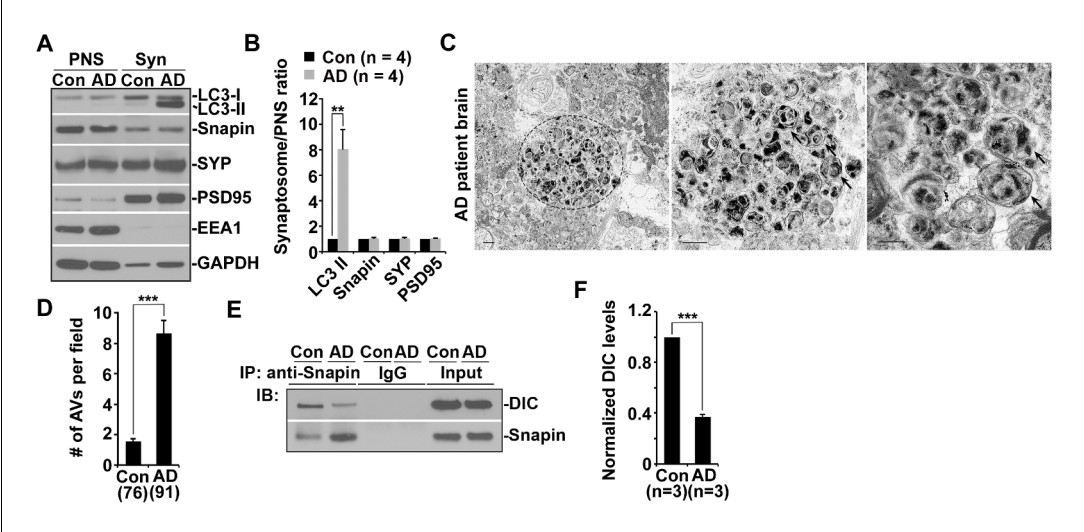

**Figure 5.** Impaired dynein-Snapin coupling contributes to axonal autophagic stress in AD patient brains. (A and B) Synaptic autophagic stress in AD patient brains. Equal amounts (15 μg) of synapse-enriched synaptosomal preparations (Syn) and post-nuclear supernatant (PNS) from human brains of control subjects and AD patients were sequentially immunoblotted on the same membrane after stripping between each antibody application. The purity of synaptosome fractions was confirmed by their relative enrichment of synaptic markers synaptophysin (SYP) and PSD95 compared to levels in PNS fractions, and by the absence of EEA1. The synaptosome/PNS ratio in AD brains were compared to those in control subjects. Data were quantified from four independent repeats. (C and D) Representative TEM images (C) and quantitative analysis (D) showing abnormal retention of AVd-like organelles (arrows) within enlarged neurites in patient brains. Note that dystrophic/swollen neurites contain predominantly late stage AVs (AVd). (E and F) Immunoprecipitation (E) and quantitative analysis (F) showing reduced Snapin-DIC coupling in AD patient brains. Data were quantified from three independent experiments. The average number of AV-like structures per EM field (10 μm × 10 μm) was quantified (D). Scale bars: 500 nm. Error bars: SEM. Student's *t* test: ***p<0.001, **p<0.01, *p<0.05.

**Table 1.** Demographic details of postmortem brain specimens from patients with AD and subjects without AD (specimens from the Harvard Tissue Resource Center and the Human Brain and Spinal Fluid Resource Center at UCLA).

| Case type | Age/sex | Postmortem interval (h) | Braak stage of AD brains |
|---|---|---|---|
| Control | 75/F | 20.1 | 0 |
| Control | 87/M | 9.3 | 0 |
| Control | 47/M | 12.5 | 0 |
| Control | 66/M | 22.5 | 0 |
| AD | 65/M | 11.6 | Braak I |
| AD | 72/M | 21.8 | Braak II |
| AD | 86/M | 9.00 | Braak III |
| AD | 86/M | 17.4 | Braak III |
| AD | 60/F | 15.2 | Braak V |
| AD | 86/F | 7.08 | Braak VI |

labeled dendrites in the hippocampal regions of *Snapin* cKO mice. Co-localized pixels of CI-MPR with MAP2 in *Snapin* cKO mice were similar to those of WT littermates (WT: 10.06 ± 2.09; *Snapin* cKO: 11.90 ± 1.17; p=0.45032), suggesting that *Snapin* deficiency results in predominant accumulation of LEs within axons negative for MAP2 (*Figure 6—figure supplement 1A,B*). Compared with the WT control, the mean intensity of CI-MPR fluorescence is significantly increased in *Snapin* cKO mouse brains (2.92 ± 0.12; p<$1 \times 10^{-16}$) (*Figure 6B*). Consistent with our previous study using cultured neurons (*Cai et al., 2010*), abnormal retention of immature lysosomes labeled by CI-MPR was also shown in the soma of the CA3 region after deletion of *Snapin* in mice (*Figure 6A*). Second, we asked whether *Snapin* deficiency results in retention of amphisomes in distal regions. We detected a significant number of AVs co-labeled with both LC3 and CI-MPR, suggesting that they had the nature of amphisomes, the late stage of AVs after fusion with LEs (*Figure 6C*). The LC3-labeled AVs clustered in the hippocampal mossy fibers of *Snapin* mutant mice (WT: 7.09 ± 1.1; *Snapin* cKO: 68.44 ± 5.43; p<$1 \times 10^{-10}$) (*Figure 6D*).

Third, we examined the recruitment of dynein motors to LEs/amphisomes by immunoisolation using Dyna magnetic beads coated with an anti-Rab7 antibody. When equal amounts of LEs/amphisomes–as reflected by Rab7 levels–were loaded, the normalized intensity of the dynein DIC in *Snapin* cKO mouse brains was significantly reduced to ~55% in comparison with that of WT littermates (p=0.003992) (*Figure 6E,F*), indicating a reduced loading of the dynein motors onto LEs/amphisomes. The significantly reduced but not fully abolished DIC recruitment in the *Snapin* cKO mouse brains may suggest (1) a compensatory role of other dynein adaptors in LE-dynein coupling, or (2) the remaining Snapin expressed in other types of cells in mouse brains. Interestingly, from the purified LEs in *Snapin* cKO mouse brains, we also detected increased LC3-II, and syntaxin 17 (Stx17) (LC3-II: p=0.0014707; Stx17: p=0.013641) (*Figure 6E,F*), an autophagosome-targeted protein mediating the fusion with late endosomes/lysosomes by forming the SNARE fusion complex with SNAP29 and VAMP8 (*Cheng et al., 2015a*; *Guo et al., 2014*; *Itakura et al., 2012*; *Wang et al., 2016*). This study further confirms that Snapin is required for dynein motor recruitment to amphisomes, and the subsequent removal of AVs from distal axons and synapses.

In addition, we performed TEM analysis to assess AV accumulation in presynaptic terminals of WT and *Snapin* cKO mice. Consistent with the results from immunostaining and immunoisolation assays, *Snapin* cKO mice exhibited a significant number of AVd-like structures at presynaptic terminals (*Figure 6G*). These AV-like organelles were not readily observed in WT synapses (WT: 0.29 ± 0.11; cKO: 1.19 ± 0.18; p<$1 \times 10^{-5}$) (*Figure 6G,H*). Moreover, massive AV accumulation within dystrophic axonal processes was also detected in *Snapin* cKO mouse brains. Compared to WT controls, the average number of AVd-like organelles per EM field within the hippocampal regions of *Snapin* cKO mice was significantly increased (WT: 0.71 ± 0.10; n = 79; *Snapin* cKO: 5.46 ± 1.0; n = 91; p<$1 \times 10^{-6}$) (*Figure 6—figure supplement 1C,D*). This observation is

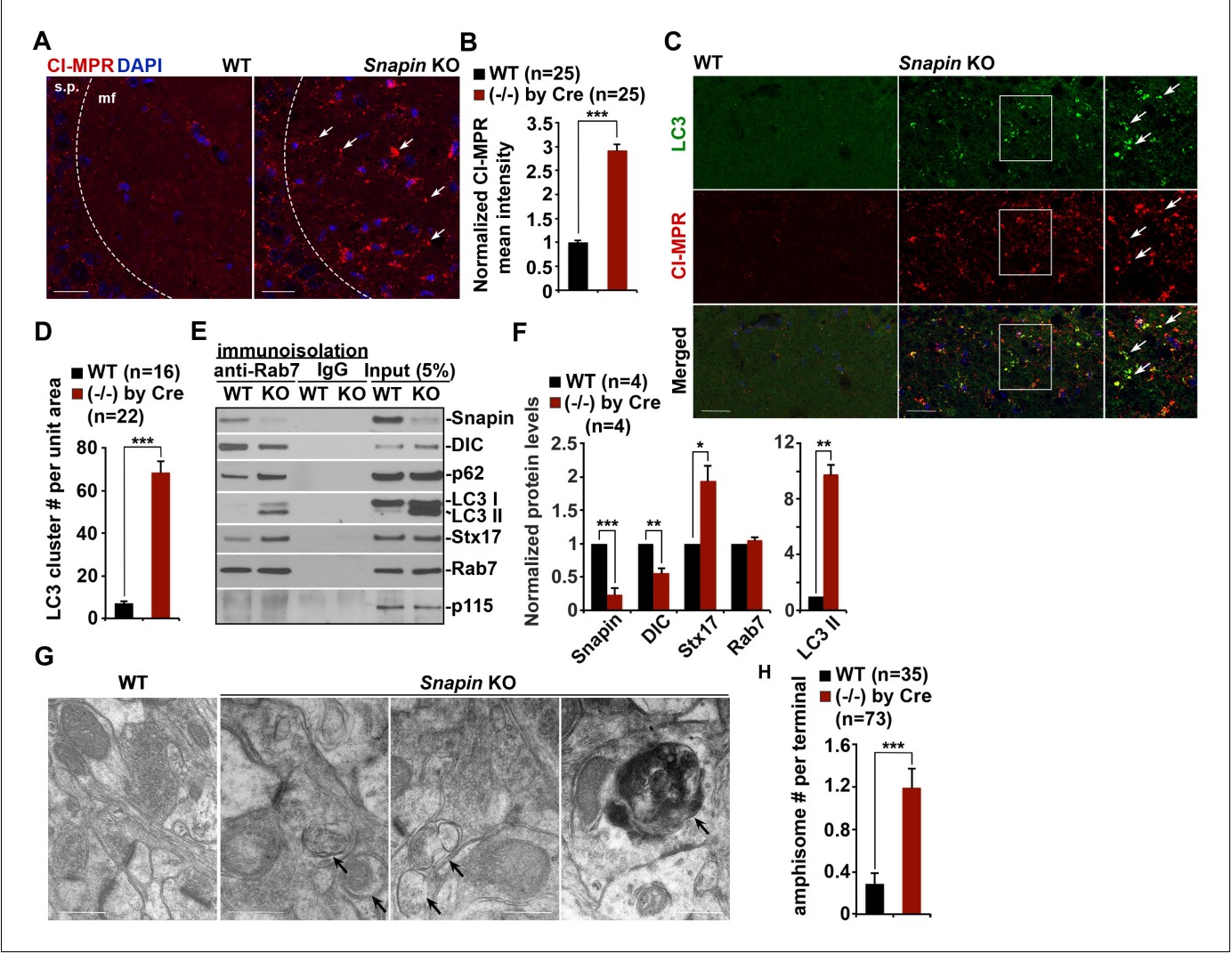

**Figure 6.** *Snapin*-deficient mouse brains recapitulate AD-associated autophagic stress in axons. (**A** and **B**) LEs clustering (arrows) in the hippocampal mossy fibers of *Snapin* flox/flox conditional knockout (cKO) mice. The mean intensity of LE clusters in the mossy fibers (mf) of one-month *Snapin* mutant mice labeled with CI-MPR per section (320 μm × 320 μm) was quantified and compared with that of WT mice. s.p., stratum pyramidale (**C** and **D**) Aberrant accumulation of amphisomes in the mossy fibers of *Snapin*-deficient mice. Note that LC3-marked AVs were labeled with CI-MPR, suggesting that they were amphisomes in nature following fusion with LEs. The number of LC3 clusters per section (320 μm × 320 μm) was quantified. (**E** and **F**) Immunoisolation showing reduced dynein attachment to amphisomes. Rab7-associated organelles were immunoisolated with anti-Rab7-coated Dyna magnetic beads, followed by sequential immunoblotting on the same membranes after stripping between each antibody application. Note that purified Rab7 organelles were enriched with various AV markers including LC3-II, p62, and syntaxin 17 (Stx17) in *Snapin* cKO mouse brains. Data were quantified from four repeats. (**G** and **H**) Representative TEM images (**G**) and quantitative analysis (**H**) showing retention of AVd-like organelles at presynaptic terminals in *Snapin* mutant mice. Arrows indicate AVd-like structures, which were not readily detected in WT mice. Scale bars: 25 μm (**A** and **C**) and 500 nm (**G**). Data were quantified from a total number of imaging slice sections (**B** and **D**) or from a total number of electron micrographs (**H**) indicated in parentheses from three pairs of mice. Error bars represent SEM. Student's *t* test: ***p<0.001, **p<0.01, *p<0.05.

The following figure supplement is available for figure 6:

**Figure supplement 1.** Axonal autophagic stress in the hippocampal regions of *Snapin*-deficient mice.

consistent with our immunostaining results (*Figure 6C,D*), suggesting autophagic stress in distal axons of *Snapin* cKO mouse brains. Thus, these morphological observations confirm that Snapin mediates recruitment of dynein motors to AVs for retrograde transport; deleting *Snapin* recapitulates AD-associated autophagic stress in axons.

# Elevated Snapin expression reduces axonal autophagic stress in mutant hAPP Tg neurons

Because *Snapin* deficiency leads to AD-like autophagic phenotypes, we sought to reverse autophagic stress by elevating Snapin expression in AD neurons. First, we assessed AV retention in AD axons. We observed reduced density of axonal AVs in hAPP neurons expressing HA-Snapin, but not the HA-Snapin-L99K mutant defective in DIC binding, or HA vector control (amphisomes per 100 μm length: 4.57 ± 0.34 for HA-Snapin, $p<1\times10^{-7}$; 8.92 ± 0.7 for HA-Snapin-L99K, p=0.50207; 8.30 ± 0.56 for HA vector) (*Figure 7A,B*). As expected, expressing HA-Snapin or HA-Snapin-L99K in mutant hAPP Tg neurons did not show significant change in the percentage of amphisomes, eliminating the possibility for defective fusion of autophagosomes with LEs by expressing Snapin (*Figure 7—figure supplement 1A*). Second, we monitored the motility of LEs and amphisomes along axonal processes. Mutant hAPP Tg neurons expressing HA-Snapin exhibited enhanced retrograde transport of both amphisomes (42.56% ± 3.04%; $p<1\times10^{-8}$) and LEs (36.36% ± 2.53%; $p<1\times10^{-8}$) along the same axons relative to control hAPP neurons expressing HA-Snapin-L99K (19.82% ± 2.97% for amphisomes; 18.35% ± 2.07% for LEs) or HA vector (17.78% ± 1.64% for amphisomes; 16.42% ± 1.40% for LEs), resulting in reduced stationary pools of both organelles (*Figure 7C–E*).

Presynaptic terminals of mutant hAPP Tg neurons exhibit AV retention (*Figure 2C,E*). We found that presynaptic autophagic accumulation was significantly reduced following overexpression of Snapin, but not Snapin-L99K in hAPP neurons (Vector: 46.69% ± 2.46%; Snapin: 18.20% ± 1.44%, $p<1\times10^{-14}$; Snapin-L99K: 51.41% ± 2.16%) (*Figure 7F,G*). We next determined whether impaired retrograde transport contributes to autophagic stress in AD axons under conditions of autophagy induction. By utilizing 10-[4′-(N-diethylamino)butyl]-2-chlorophenoxazine (10-NCP), which induces neuronal autophagy in an mTOR-independent fashion (*Tsvetkov et al., 2010*), we examined whether Snapin-mediated enhancement of AV transport rescues autophagic stress upon autophagy induction. Axonal AV density was increased in hAPP neurons treated with 10-NCP relative to untreated controls (AVs per 100 μm length: hAPP 8.87 ± 0.4; hAPP with 10-NCP 21.26 ± 1.12; $p<1\times10^{-12}$) (*Figure 7—figure supplement 1B,C*). 10-NCP-induced autophagy did not lead to the increase in the axonal AV density of AD neurons overexpressing Snapin (AVs per 100 μm length: 4.71 ± 0.31; $p<1\times10^{-11}$) as a result of enhanced retrograde transport (10-NCP treated hAPP neurons: 11.3% ± 1.19%; hAPP neurons expressing Snapin: 43.57% ± 3.4%, $p<1\times10^{-8}$) (*Figure 7—figure supplement 1D,E*). These results suggest that enhanced retrograde transport of AVs by elevated Snapin expression efficiently removes AVs from distal axons and presynaptic terminals, thus reducing autophagic stress in AD axons. Altogether, our study provides new mechanistic insights into AD-associated axonal autophagic stress, establishing a foundation for ameliorating axonal pathology in AD.

## Discussion

Defective autophagy has been implicated in AD pathogenesis (*Funderburk et al., 2010*; *Nixon, 2013*; *Nixon and Yang, 2011*). The presence of massively accumulated AVs in dystrophic (swollen) neurites is a unique feature linked to AD pathology (*Nixon et al., 2005*). Microtubule-based long-distance axonal transport is essential for autophagic clearance because autophagosomes are predominantly generated in distal axons and rely heavily on retrograde transport toward the soma for lysosomal proteolysis (*Cheng et al., 2015a*, *2015b*; *Lee et al., 2011a*; *Maday and Holzbaur, 2016*; *Maday et al., 2012*). Such a mechanism enables neurons to efficiently remove autophagic cargos from axons and synapses, thus reducing autophagic stress. While previous studies provide important information about autophagic flux and trafficking in healthy neurons under physiological conditions, mechanisms underlying AD-linked autophagic stress remain largely unknown.

In the current study, we provide mechanistic insights into AD-associated autophagic stress in AD neurons and patient brains. First, we reveal that amphisomes predominantly accumulate in distal axons and at the presynaptic terminals of mutant hAPP Tg mouse brains (*Figure 1* and *Figure 1—figure supplement 1*). Second, we show that retrograde transport of amphisomes is impaired in mutant hAPP Tg neurons, leading to axonal and presynaptic retention of AVs (*Figure 2* and *Figure 2—figure supplement 1*). Third, we reveal that cytoplasmic Aβ1-42 oligomers associates with these accumulated amphisomes in the distal axons of AD mice (*Figure 3* and *Figure 3—figure*

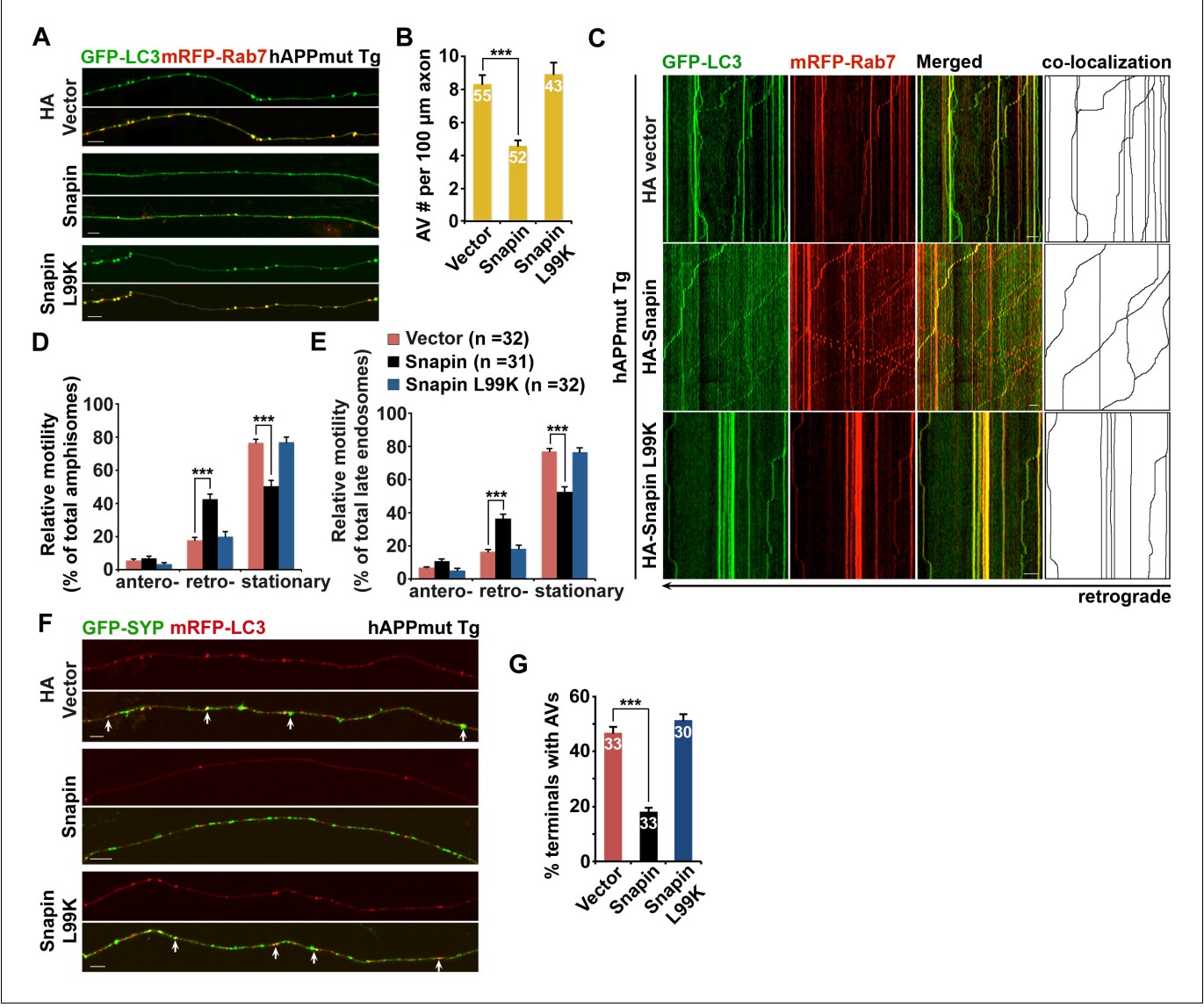

**Figure 7.** Elevated Snapin expression reduces axonal autophagic stress of mutant hAPP Tg neurons. (A and B) Images (A) and quantitative analysis (B) showing that expressing Snapin, but not the Snapin-L99K mutant, reduces the density of axonal amphisomes in mutant hAPP Tg neurons. (C–E) Kymographs (C) and quantitative analysis (D and E) showing that impaired amphisome retrograde transport was rescued by expressing Snapin, but not Snapin-L99K. hAPP neurons were co-transfected with GFP-LC3 and mRFP-Rab7 along with HA-Snapin, HA-Snapin-L99K, or HA vector, followed by time-lapse imaging at DIV17-19. (F and G) Representative images (F) and quantitative analysis (G) showing reduced autophagic accumulation at presynaptic terminals after overexpression of Snapin, but not Snapin-L99K in hAPP neurons. Scale bars: 5 μm (A) and 10 μm (C and F). Data were quantified from a total number of neurons (n) indicated on the top of bars (B and G) or in parentheses (D and E) from more than four independent experiments. Error bars: SEM. Student's *t* test: ***p<0.001, **p<0.01, *p<0.05.

The following figure supplement is available for figure 7:

**Figure supplement 1.** Snapin-mediated rescue effects on axonal autophagic stress under conditions of autophagy induction in AD neurons.

supplement 1). Fourth, the Aβ1-42-DIC interaction interferes with dynein-Snapin coupling and reduces the recruitment of dynein motor to amphisomes, thus impairing AV transport (*Figure 4*). Fifth, we further confirm such impaired dynein-Snapin coupling in AD patient brains (*Figure 5*). Sixth, we demonstrate that *Snapin* deficiency in mice recapitulates axonal autophagic stress (*Figure 6* and *Figure 6—figure supplement 1*). Seventh, we show that elevated Snapin expression in AD neurons reduces axonal AV retention by enhancing their retrograde transport (*Figure 7* and *Figure 7—figure*

*supplement 1*). Therefore, our study provides the first indication that defects in dynein-Snapin-driven AV retrograde transport contributes to AD-associated axonal autophagic stress.

Under AD-associated pathological conditions, both endocytic and autophagic pathways are sites of APP processing and Aβ production (*Nixon, 2007*). It was reported that Aβ peptides oligomerize and accumulate within neuronal processes in AD mice and patient brains (*Takahashi et al., 2004*). This raises a fundamental question as to whether overloaded Aβ in axons impairs dynein-driven retrograde transport of axonal cargoes such as LEs and AVs, two main organelles in the endo-lysosomal and autophagic pathways. In this study, we showed that soluble Aβ associates with amphisomes in the distal AD axons of AD neurons (*Figure 3*). Through direct interaction with DIC, Aβ1-42 interferes with the assembly of dynein-Snapin motor-adaptor complex, thus interrupting the recruitment of dynein motors to LEs and amphisomes for driving their retrograde transport (*Figure 4*). Given that AD-linked autophagic stress is uniquely associated with Aβ generation (*Benzing et al., 1993*), our findings suggest that dynein is a target of Aβ-mediated toxicity, thus resulting in impaired dynein-Snapin coupling and AV retrograde transport in AD axons. *Snapin* mutant mouse brains exhibit a striking phenotype: AD-like axonal autophagic stress (*Figure 6* and *Figure 6—figure supplement 1*). More importantly, elevated Snapin expression reverses AV retention by enhancing AV retrograde transport in the axons and presynaptic terminals of AD neurons (*Figure 7* and *Figure 7—figure supplement 1*). These results are consistent with a recent study showing the in vitro and in vivo rescue effects of enhanced Snapin expression on the clearance of autophagic cargos in the axons of amyotrophic lateral sclerosis (ALS)-linked motor neurons (*Xie et al., 2015*). Therefore, our results support a model that impaired AV retrograde transport plays a critical role in autophagic pathology in AD axons.

Many studies have been focused on the mechanisms underlying defects in kinesin-mediated anterograde transport in AD (*Pigino et al., 2009, 2003*; *Stokin et al., 2005*; *Tang et al., 2012*). Synthetic Aβ1-42 was proposed to inhibit axonal transport through CK2 activation, which regulates kinesin-1 light chain and thus cargo attachment of kinesin-1, the anterograde transport motors (*Pigino et al., 2009*). A recent study showed that cell-derived soluble Aβ-oligomers induced early and selective diminutions in anterograde transport of synaptic cargoes (*Tang et al., 2012*). These reported toxic effects on axonal transport were elicited by exogenously added Aβ, and were proposed to depend on CK2, GSK3, or tau (*Decker et al., 2010*; *Pigino et al., 2009, 2003*; *Rui et al., 2006*; *Stokin et al., 2005*; *Tang et al., 2012*; *Vossel et al., 2010*). To our knowledge, it has not been investigated whether intracellular Aβ impairs dynein motor-driven axonal transport, in particular, whether and how intracellular Aβ disrupts Snapin-DIC interaction and thus AV retrograde transport, and whether these defects contribute to autophagic pathology in AD.

We showed that soluble Aβ1-42 oligomers are enriched within distal axons and presynaptic terminals of AD mice, where they associate with accumulated amphisomes (*Figure 3—figure supplement 1*). In our immuno-EM analysis, we provided more direct evidence showing the presence of Aβ outside of the AVs. The immuno-gold anti-A11 antibody-labeled oligomeric Aβ is mostly present in the cytoplasm and is enriched in AD axons (*Figure 3F,G,I*). More importantly, a majority of these cytoplasmic Aβ gold particles are located outside of AVs and associate with or surround AVs (*Figure 3H, I*). Thus, we proposed that soluble Aβ oligomers in the cytoplasm interfere with the assembly of the DIC-Snapin complex through direct interaction with dynein DIC, thus interrupting cytoplasmic dynein motor recruitment to Snapin-associated AVs. We showed that this interruption was detected in the concentration of oligomeric Aβ1-42 as low as 0.2 μM in vitro in our study (*Figure 4*). These findings suggest that dynein-Snapin coupling, and thus cargo-motor association, is compromised in response to cytoplasmic accumulation of Aβ1-42 oligomers, which could be a pathogenic mechanism of impaired dynein-driven retrograde transport in AD.

Recent studies from several groups provide consistent evidence that autophagosomes are predominantly generated in distal axons, and undergo retrograde transport toward the soma for lysosomal proteolysis (*Cheng et al., 2015a*; *Fu et al., 2014*; *Lee et al., 2011a*; *Maday and Holzbaur, 2014*; *Maday et al., 2012*). One recent study reported that newly generated autophagosomes in distal axon undergo robust retrograde transport toward the soma for maturation into autolysosomes (*Maday and Holzbaur, 2016*). They showed that these axon-generated autophagosomes enter the soma and then remain in the somatodendritic domain. Such compartmentalization facilitates degradation of axonal AVs in the soma, where mature lysosomes are mainly located. Consistently, the density of autophagosomes in the soma, but not in the axon, was increased by blocking lysosome

function with Bafilomycin A1. Their study concluded that lysosomal proteolysis in the soma is essential for the clearance of autophagic cargoes, which are mostly delivered from distal axons (*Maday and Holzbaur, 2016*). This study also showed that some autophagosomes are locally formed in the soma, are less motile than the axonal ones under basal condition, and do not increase in response to stress induced autophagy by nutrient deprivation. Together, this study and others consistently support our current findings that the soma is the primary site for autophagosome clearance; axonal autophagosomes undergo retrograde transport from the distal region to the soma for their degradation in order to maintain axonal homeostasis (*Cai et al., 2010*; *Cheng et al., 2015a*; *Lee et al., 2011a*; *Maday and Holzbaur, 2014*; *Maday et al., 2012*).

We have focused on the mechanism underlying autophagic pathology in the axon, but not in the soma of AD neurons because AD brains exhibit a unique phenotype—massive accumulation of AVs within distal dystrophic neurites (*Figure 1* and *Figure 5*). Our study addresses a new fundamental issue that retrograde transport is critical for the reduction of autophagic stress in distal axons under AD condition. (1) AD axons display predominant amphisome accumulation and reduced retrograde transport of LEs/amphisomes (*Figure 2* and *Figure 2—figure supplement 1*); (2) Expression Snapin, but not its dynein-binding defective mutant, significantly reduces AV retention by enhancing their retrograde transport in AD axons (*Figure 7*); (3) Snapin-enhanced AV transport rescues AV accumulation in AD axons upon autophagy induction (*Figure 7—figure supplement 1*); and (4) deleting *Snapin* recapitulates AD-associated autophagic stress in distal axons (*Figure 6* and *Figure 6—figure supplement 1*). Given that the soma is the primary site of AV clearance (*Maday and Holzbaur, 2016*), our data consistently suggests that impeded AV retrograde transport contributes to autophagic pathology in AD axons.

*Martinez-Vicente et al. (2010)* revealed an important mechanism that inefficient autophagy is attributed to cargo recognition failure in Huntington's disease. However, there is no indication of a similar mechanism in AD (*Nixon, 2013*). Instead, amphisomes/autolysosmes, rather than empty autophagic organelles, are the predominant AVs accumulated in AD brains (*Nixon, 2007*; *Nixon et al., 2005*; *Yu et al., 2005*). We provide new evidence showing aberrant accumulation of AVd-like structures containing vesicles and partially digested materials accumulate in distal axons and presynaptic terminals of AD mouse model and patient brains (*Figure 1*, *Figure 2*, *Figure 3*, and *Figure 5*). Moreover, these AVs contain ubiquitinated cargoes (*Figure 1—figure supplement 1*, *Figure 2—figure supplement 1*, and *Figure 3—figure supplement 1*). In addition, deleting *Snapin* in mice, which impedes AV transport, but does not affect cargo engulfment, exhibits similar phenotypes of autophagic accumulation (*Figure 6* and *Figure 6—figure supplement 1*). These findings suggest that cargo recognition failure in AD is unlikely a predominant mechanism underlying AD-associated autophagic stress.

Proper lysosomal function is critical for the removal of autophagic substrates. Autophagic stress has been linked to lysosomal storage diseases (LSDs) (*de Pablo-Latorre et al., 2012*; *Sano et al., 2009*). The lysosomal deficits in AD are thought to play a critical role in autophagy dysfunction, leading to the disruption of substrate proteolysis within autolysosomes (*Boland et al., 2008*; *Lee et al., 2011b*; *Yang et al., 2011*). Defective lysosomal proteolysis produces similar neuropathology in WT mice and exacerbates amyloid and autophagy pathology in AD mouse models (*Nixon, 2013*; *Nixon and Yang, 2011*). Because neurons are particularly dependent on lysosomal degradation capacity for eliminating A$\beta$ generated in the endocytic-autophagic pathways (*LeBlanc and Goodyer, 1999*; *Lee et al., 2011a*), impaired AV retrograde transport we observed in AD neurons could be indirectly attributed to lysosomal deficits. Thus, increased A$\beta$ load upon lysosomal inhibition may impair the recruitment of dynein motors to AVs by competitively interfering with dynein-Snapin coupling in AD neurons.

In summary, our study provides new mechanistic insights into autophagic defects under AD-linked pathogenesis, which conceptually advances current knowledge of A$\beta$-induced impairment of dynein-mediated retrograde transport underlying autophagic stress in axons and at synapses of AD brains. Given that synaptic retention of AVs alters synaptic structure and neurotransmission, elucidation of this pathological mechanism has a broad neurobiological impact because impaired axonal transport, autophagic stress, and synaptic dysfunction are all associated with major neurodegenerative diseases. Therefore, enhancing clearance of AVs by regulating retrograde trafficking may be a potential therapeutic strategy for AD and other major neurodegenerative diseases. This study advances our understanding of autophagic stress in AD and may have broader relevance to other

neurodegenerative diseases associated with defective axonal transport and autophagy dysfunction. Further therapeutic approaches aimed at regulating the dynein-Snapin coupling may help attenuate axonal pathology in AD.

## Material and methods

### Mice

*Snapin flox* mice were provided by ZH Sheng (National Institute of Neurologic Disorders and Stroke, NIH, Bethesda, MD). *Camk2α*-tTA and tet-APP[swe/ind] mice were obtained from H Cai (National Institute on Aging, NIH, Bethesda, MD). hAPP mice (C57BL/6J) from line J20 (*Mucke et al., 2000*) and *Thy1*-Cre Tg mice (*Campsall et al., 2002*) were purchased from the Jackson Laboratory (Bar Harbor, ME).

### Human brain specimens

Postmortem brain specimens from AD patients and age-matched control subjects were obtained from the Harvard Tissue Resource Center, and the Human Brain and Spinal Fluid Resource Center at UCLA. Specimens were from patients diagnosed with AD according to Braak criteria (*Braak and Braak, 1991*). The specimens were from the frontal cortex and were quick-frozen (BA9) (*Table 1*). The EM data were from two control subjects and three AD patient brains at different Braak stages with postmortem interval between 7.08 hr and 12.5 hr. Four control subjects and four patient brains (postmortem interval 7.08 hr – 22.5 hr) were used for synaptosomal fraction purification. The data of immunoprecipitation assays were from three sets of human brains with postmortem interval 9.00 hr – 20.1 hr.

### Materials

pmRFP-Rab7 and pmRFP-Ub were from A. Helenius and N. Dantuma, respectively. The constructs encoding Snapin, Snapin-L99K, GFP-DIC, GST-DIC, GFP-LC3, hAPP, and hAPP[swe], were prepared as previously described (*Cai et al., 2010*; *Ye and Cai, 2014*; *Zhou et al., 2012*). The purified polyclonal antibody against mouse N-terminal Snapin was described previously (*Tian et al., 2005*) and obtained from Z.H. Sheng. Sources of other antibodies and reagents are as follows: polyclonal anti-EEA1, anti-MAP2, anti-syntaxin 1, anti-TOM20, and anti-synaptophysin antibodies (Santa Cruz, Dallas, TX); monoclonal anti-DIC, anti-GAPDH, and anti-synaptophysin antibodies, polyclonal Aβ1-42 and anti-APP c-terminal antibodies (Millipore/CHEMICON, Billerica, MA); monoclonal anti-GFP (JL-8) antibody (Clontech, Madison, WI); monoclonal anti-β Amyloid (6E10) and anti-HA antibodies (Biolegend, San Diego, CA); monoclonal anti-p115, anti-MAP2, and anti-p150[Glued] antibodies (BD Biosciences, San Jose, CA); monoclonal anti-Ubiquitin and polyclonal anti-LC3 antibodies (Cell Signaling Technology, Danvers, MA); monoclonal anti-p62/SQSTM1 antibody (Abnova, Taiwan, China); monoclonal anti-Rab7 and anti-Neurofilament 200 antibodies, and polyclonal anti-LC3, anti-Neurofilament 160, and anti-syntaxin 17 antibodies (Sigma, St. Louis, MO); monoclonal anti-GST antibody (Thermo scientific, Grand Island, NY); monoclonal anti-PSD95 antibody (UpstateMillipore/CHEMICONUpstate Upstate, Billerica, MA); monoclonal anti-CI-MPR and anti-LAMP-1 antibodies were developed by D Messner and JT August and were obtained from Developmental Studies Hybridoma Bank (Iowa City, Iowa). Polyclonal anti-Aβ (A11) antibody and Alexa fluor 546-, and 633-conjugated secondary antibodies (Invitrogen/Thermo scientific, Grand Island, NY). COS7 Cells were purchased from ATCC (CRL-1651) (Manassas, VA ). The authenticity was confirmed by STR profiling. The mycoplasma contamination was tested and showed negative result prior to the experiments.

### Transfection and immunocytochemistry of cultured cortical neurons

Cortices were dissected from E18–19 mouse embryos as described (*Cai et al., 2010*, *2012*; *Goslin and Banker, 1998*). Cortical neurons were dissociated by papain (Worthington, Lakewood, NJ) and plated at a density of 100,000 cells per cm$^2$ on polyornithine- and fibronectin-coated coverslips. Neurons were grown overnight in plating medium (5% FBS, insulin, glutamate, G5 and B27) supplemented with $100 \times$ L-glutamine in Neurobasal medium (Invitrogen). Starting at DIV 2, cultures were maintained in conditioned medium with half-feed changes of neuronal feed (B27 in Neurobasal medium) every 3 days. Primary hAPP Tg neurons were cultured from

breeding mice of hemizygous mutant hAPP[Swe/Ind] Tg (J20 line) with WT animals (*Mucke et al., 2000*). Genotyping assays were performed following culture plating to verify mouse genotypes. In our study, we examined both transgenic neurons and non-transgenic neurons derived from their littermates. Neurons were transfected with various constructs using Lipofectamine 2000 (Invitrogen) followed by time-lapse imaging at DIV16-20 transfection prior to qualification analysis.

## COS7 cell culture and transfection
COS7 Cells (ATCC) were incubated with high glucose DMEM containing sodium pyruvate, L-glutamine, supplemented with 10% FBS and penicillin-streptomycin (Invitrogen). Transient transfection COS7 was performed using Lipofectamine 2000. 100 μl of Opti-MEM (Invitrogen) and 1–2 μl of Lipofectamine 2000 (Invitrogen) per chamber were pre-incubated at room temperature (RT) for 5 min and then mixed with 100 μl of Opti-MEM containing DNA constructs (2–3 μg per chamber) and incubated for 20 min at RT to allow complex formation. The entire mixture was added directly to cultured cells. Following transfection, cells were cultured for an additional 1–2 days before harvesting for biochemical analysis.

## Tissue preparation and immunohistochemistry
Animals were anaesthetized with 2.5% avertin (0.5 ml per mouse), and transcardially perfused with fixation buffer (4% paraformaldehyde in PBS, pH 7.4). Brains were dissected out and postfixed in fixation buffer overnight and then placed in 30% sucrose at 4°C. 10-μm-thick coronal sections were collected consecutively to the level of the hippocampus and used to study co-localization of various markers. After incubation with blocking buffer (2.5% goat serum, 0.15% Triton X-100, 1.5% BSA, 0.5% glycine in $H_2O$) at RT for 1 hr, the sections were incubated with primary antibodies at 4°C overnight, followed by incubating with secondary fluorescence antibodies at 1:400 dilution at RT for 1 hr. After fluorescence immunolabeling, the sections were stained with DAPI, washed three times in PBS. The sections were then mounted with anti-fading medium (vector laboratories, H-5000, UK) for imaging. Confocal images were obtained using an Olympus FV1000 oil immersion 40 × objective with sequential-acquisition setting. Eight to ten sections were taken from top-to-bottom of the specimen and brightest point projections were made.

## Quantification of co-localization
A threshold intensity was determined in the thresholding function of NIH Image J, which was preset for the fluorescent signals as previously described (*Nagahara et al., 2013*). The co-localized pixels above the threshold intensity were automatically quantified and scored by Image J based on the fluorescence intensity profile, which was expressed as co-localized mean intensity positive for both channels. The co-localization was presented as the percentage of the co-localized intensity relative to total fluorescence intensity.

## Image acquisition and quantification
Confocal images were obtained using an Olympus FV1000 oil immersion 60 × objective (1.3 numerical aperture) with sequential-acquisition setting. For fluorescent quantification, images were acquired using the same settings below saturation at a resolution of 1024 × 1024 pixels (8 bit). Eight to ten sections were taken from top-to-bottom of the specimen and brightest point projections were made. Morphometric measurements were performed using NIH ImageJ. Measured data were imported into Excel software for analysis. The thresholds in all images were set to similar levels. Data were obtained from at least three independent experiments and the number of cells or imaging sections used for quantification is indicated in the figures. All statistical analyses were performed using the Student's *t*-test and are presented as mean ± SEM.

For live cell imaging, cells were transferred to Tyrode's solution containing 10 mM Hepes, 10 mM glucose, 1.2 mM $CaCl_2$, 1.2 mM $MgCl_2$, 3 mM KCl and 145 mM NaCl, pH 7.4. Temperature was maintained at 37°C with an air stream incubator. Cells were visualized with a 60 × oil immersion lens (1.3 numerical aperture) on an Olympus FV1000 confocal microscope, using 488 nm excitation for GFP and 543 nm for mRFP. Time-lapse sequences of 1024 × 1024 pixels (8 bit) were collected at 1–2 s intervals with 1% intensity of the argon laser to minimize laser-induced bleaching and damage to cells, and maximum pinhole opening. Dual-color time-lapse images were captured

by a total of 100 frames. All recordings started 6 min after the coverslip was placed in the chamber. The stacks of representative images were imported into NIH ImageJ. A membranous organelle was considered stopped if it remained stationary for the entire recording period; a motile one was counted only if the displacement was at least 5 μm.

## Criteria for axon selection in cultured neurons

For analyzing the motility of AVs or late endosomes in live neurons, we selected axons for time-lapse imaging and measuring motility because axons, but not dendrites, have a uniform microtubule organization and polarity. Axonal processes were selected as we previously reported (*Cai et al., 2010*, *2012*; *Kang et al., 2008*). Briefly, axons in live images were distinguished from dendrites based on known morphological characteristics: greater length, thin and uniform diameter, and sparse branching (*Banker and Cowan, 1979*). Only those that appeared to be single axons and separate from other processes in the field were chosen for recording axonal AVs or late endosomes transport. Regions where crossing or fasciculation occurred were excluded from analysis.

Kymographs were used to trace axonal anterograde or retrograde movement of membranous organelles and to count stationary ones as described previously (*Kang et al., 2008*; *Miller and Sheetz, 2004*) with extra plug-ins for ImageJ (NIH). Briefly, we used the 'Straighten' plugin to straighten curved axons and the 'Grouped ZProjector' to z-axially project re-sliced time-lapse images. The height of the kymographs represents recording time (100 s unless otherwise noted), while the width represents the length (μm) of the axon imaged. Counts were averaged from 100 frames for each time-lapse image to ensure accuracy of stationary and motile events. Measurements are presented as mean ± SEM. Statistical analyses were performed using unpaired Student's *t*-tests.

## Preparation of Synapse-enriched fractions

Synaptosome preparations from WT and mutant hAPP Tg mouse brains or AD patient brains and control subjects were collected using Percoll gradient centrifugation as described in the protocol (*Leenders et al., 2004*). Cortex tissues were homogenized in ice cold Sucrose Buffer [5 mM HEPES, 1 mM EDTA, 0.32 M sucrose and protease inhibitors (Roche, Indianapolis, IN), pH 7.4]. Homogenates were centrifuged at 1000 × g for 10 min, the supernatant was gathered and overlaid on Percoll gradients that has 2 ml of 10% Percoll gradient layered over 15%, 23%, and 40% Percoll gradients. The gradient was then separated by centrifugation for 5 min at 32,500 × g. The synaptosomal fraction was collected from the 15%/23% Percoll layers, and combined with 5 ml the Sucrose buffer. The mixture was then centrifuged at 15,000 × g for 15 min and resuspended in the Sucrose buffer. Protein quantification was performed by BCA assay (Pierce Chemical Co./Thermo scientific). 15 μg of protein from synaptosome and post nuclear supernatant (PNS) homogenates were resolved by 4–12% SDS-PAGE for sequential Western blots on the same membranes after stripping between each application of antibody. For multiple detection with different antibodies, blots were first stripped in a solution of 62.5 mM Tris-HCl, pH 7.5, 20 mM dithiothreitol and 1% SDS for 15 min at 50°C with agitation and then washed with TBS/0.1% Tween-20 for 2 × 15 min (*Leenders et al., 2004*; *Cai et al., 2010*).

## Preparation of light membrane fraction (LMF) and immunoisolation of late endocytic organelles

Brain tissues from WT or mutant hAPP Tg or *Snapin* cKO mice were homogenized in the buffer (10 mM HEPES [pH 7.4], 1 mM EDTA, 0.25 M sucrose, and protease inhibitors) and centrifuged at 800 × g for 10 min, and the supernatant was collected. The pellet was re-suspended in the homogenization buffer using a glass rod with 3 to 4 gentle strokes of the pestle of the 30 ml Dounce Homogenizer and re-centrifuged at 800 × g for 10 min. The combined first and second supernatants were centrifuged at 3500 × g for 10 min and then collected for high-speed centrifugation at 20,000 × g for 10 min. The pellet was re-suspended in the homogenization buffer using a glass rod with 3 to 4 gentle strokes of the pestle of the 30 ml Dounce Homogenizer and re-centrifuged at 20,000 × g for 10 min. The pellet was then re-suspended in the homogenization buffer as light membrane fraction (LMF) and subjected to immuno-isolation with tosylated linker-coated superparamagnetic beads (Dynabeads M-450 Subcellular; Invitrogen) as previously described (*Cai et al., 2010*; *Ye and Cai, 2014*; *Zhou et al., 2012*). For all subsequent steps, beads were collected with a magnetic device (MPC;

Invitrogen). After washing once for 5 min in PBS (pH 7.4) with 0.1% BSA at 4°C, the linker-coated beads (1.4 mg) were incubated with 1 μg anti-Rab7 mAb, or control mouse IgG overnight at 4°C on a rotator. After incubation, the beads were washed four times (5 min each) in PBS [pH 7.4] with 0.1% BSA at 4°C, and then re-suspended in an incubation buffer containing PBS [pH 7.4], 2 mM EDTA, 5% fetal bovine serum. Approximately 400 μg of light membrane fraction from mutant hAPP Tg or *Snapin* cKO mouse brains were mixed with incubation buffer containing beads (final reaction volume 1 ml) and incubated for 4 hr at 4°C on a rotator. After incubation, the beads were collected with a magnetic device and washed five times with the incubation buffer and three times with PBS for 10 min each and then resolved by 4–12% Bis-Tris PAGE for sequential Western blots on the same membranes after stripping between each application of the antibody. For semi-quantitative analysis, protein bands detected by ECL were scanned into Adobe Photoshop CS6, and analyzed using NIH ImageJ.

## Dynein intermediate chain Pull-down assay

Dynein intermediate chain (DIC) pull-down experiments were performed as previously described [77]. In brief, GST fusion proteins were bound to glutathione-Sepharose beads (GE Healthcare, Port Washington, NY) in TBS buffer (50 mM Tris-HCl at pH 7.5, 140 mM NaCl) with 0.1% Triton X-100 and protease inhibitors, and incubated at 4°C for 1 hr with constant agitation, followed by washing for three time with TBS. The beads coupled with ~1 μg GST fusion protein were added to 2 mg LMF prepared from mouse brains in the presence or absence of Aβ1-42, and then incubated overnight at 4°C. The beads were then washed three times with TBS buffer; bound proteins were processed for 4–12% Bis-Tris PAGE and immunoblotting on the same membranes after stripping between each application of the antibody.

## Preparation of oligomeric Aβ

Scrambled Aβ, oligomeric Aβ1-42, or Aβ1-40 was prepared as previously described (*Du et al., 2010*; *Rui et al., 2006*; *Vossel et al., 2010*). Aβ1-42 or Aβ1-40 peptide (Sigma) was diluted in 1,1,1,3,3,3-hexafluoro-2-propanol to 1 mM using a glass gas-tight Hamilton syringe with a Teflon plunger. The clear solution was then aliquoted in microcentrifuge tubes, followed by evaporation in the fume hood over night at RT, and it was then dried under vacuum for 1 hr in a speedVac (DNA Vap, Labnet, Edison, NJ). Peptide film was diluted in DMSO to 5 mM and sonicated for 10 min in bath sonicator. The peptide solution was resuspended in cold TBS buffer to 100 μM and immediately vortexed for 30 s; the solution containing monomeric Aβ was then incubated at 4°C for 24 hr to form oligomeric Aβ before applying to in vitro binding assays.

## Fusion protein preparation, in vitro binding, and Immunoprecipitation

DIC and Snapin were constructed into the GST-fusion vector pGEX-4T (GE Healthcare) and the His-tagged vector pET28a (Novagen/EMD, Billerica, MA), respectively. Fusion proteins were prepared as crude bacterial lysates by mild sonication in PBS containing 1% Triton X-100 and protease inhibitors (1 mM phenylmethylsulfonyl fluoride, 10 μg/ml leupeptin, 2 μg/ml aprotinin). in vitro binding experiments were performed as described previously (*Cai et al., 2010*). In brief, GST fusion proteins were bound to glutathione-Sepharose beads (GE Healthcare) in TBS buffer (50 mM Tris-HCl at pH 7.5, 140 mM NaCl) with 0.1% Triton X-100 and protease inhibitors, incubated at 4°C for 1 hr with constant agitation and washed with TBS. The beads coupled with ~1 μg GST fusion protein were added to Aβ or His-Snapin lysates, and then incubated for 3 hr at 4°C. The beads were washed three times with TBS; bound proteins were processed for 4–12% Bis-Tris PAGE and immunoblotting on the same membranes after stripping between each application of the antibody, or spotted for the dot blot analysis to detect Aβ.

For immunoprecipitation, an equal amount (600 μg) of COS7 cell lysates or (750 μg) human brain homogenates from AD patients or control subjects or (750 μg) brain homogenates from WT and mutant hAPP Tg mice were incubated with anti-GFP or anti-Snapin or anti-DIC antibody in 200 μl of TBS with 0.1% Triton X-100 and protease inhibitors, and incubated on a rotator at 4°C for overnight. 2.5 mg Protein A-Sepharose CL-4B resin (GE Healthcare) were added to each sample, and the incubation continued for an additional 3 hr followed by three washes with TBS/0.1% Triton X-100.

Immobilized protein complexes were processed for 4–12% Bis-Tris PAGE and immunoblotting on the same membranes after stripping between each application of the antibody.

For semi-quantitative analysis, protein bands detected by ECL were scanned into Adobe Photoshop CS6 and analyzed using NIH ImageJ. Care was taken during exposure of the ECL film to ensure that intensity readouts were in a linear range of standard curve blot detected by the same antibody. Paired Student $t$-tests were carried out and results are expressed as mean ± SEM.

### Transmission electron microscopy

Hippocampi from WT and mutant hAPP Tg mice, or AD patient brains were cut into small specimens (one dimension <1 mm) and fixed in Trumps fixative (Electron Microscopy Sciences, Hatfield, PA) for 2 hr at RT. The sections were then washed by 0.1 M Cacodylate buffer, and postfixed in 1% osmium tetroxide, followed by dehydrating in ethanol, and embedding using the EM bed 812 kit (Electron Microscopy Sciences) according to a stand procedure. Cultured mouse cortical neurons at DIV18-19 were fixed at RT with EM fixative (2% glutaraldehyde and 2% paraformaldehyde in 0.1 N Na$^+$ cacodylate buffer) for 2 hr. Samples were then stored at 4°C for overnight and then treated with osmium tetroxide, en bloc mordanted with uranyl acetate, dehydrated through a series of graded ethanol washes, and embedded in epoxy resins. Images were acquired on an electron microscope (100C ×; JEOL) (Division of Life Sciences, Rutgers University Electron Imaging Facility). For quantitative studies, AVs were characterized by initial AVs (AVi) with double-membrane structures containing organelles or vesicles, or late-stage degradative AVs (AVd) after fusion with late endocytic organelles containing partially degraded cytoplasmic material, small vesicles, and electron-dense material (*Cheng et al., 2015a*; *Klionsky et al., 2012*). Quantification analysis was performed blindly to condition.

Immuno-EM were performed as described previously (*Cai et al., 2010*). In brief, cultured cortical neurons at DIV19-20 were fixed with 4% paraformaldehyde and 0.05% glutaraldehyde in PBS for 45 min, washed, and then permeabilized and blocked with 0.1% saponin/5% normal goat serum in PBS for 1 hr, incubated with primary antibody for 1 hr at RT, washed, and incubated with secondary antibody conjugated to 1.4 nm Nanogold (Nanoprobes, Yaphank, NY) for 1 hr. Samples were fixed with 2% glutaraldehyde in PBS, washed and followed by silver enhancement (Nanoprobes) for 15 min, treated sequentially with 0.2% OsO4 in phosphate buffer for 30 min and 0.25% uranyl acetate at 4°C overnight, washed and dehydrated in ethanol and finally embedded in epoxy resins. All steps were carried out at RT unless otherwise indicated. The control for specificity of immunolabeling omitted the primary antibody.

## Acknowledgements

We thank ZH Sheng at NINDS/NIH for important reagents and *Snapin* mice; S Cheng at NINDS EM facility, Valentin Starovoytov at EM facility in the Department of Cell Biology and Neuroscience, and Rajesh Patel at EM facility in the Department of pathology and Laboratory Medicine, Robert Wood Johnson Medical School for technical help; E Gavin, D Ling, J Filtes, R Pillai, J Sheu, J Lam and other members in QC lab for their research assistance; Harvard Tissue Resource Center supported by National Institutes of Health grant HHSN-271-2013-00030C and Human Brain and Spinal Fluid Resource Center at UCLA for providing the postmortem brain specimens from AD patients and age-matched control subjects. This research was supported by the National Institutes of Health [R00AG033658 and R01NS089737 to QC]; the Alzheimer's Association [NIRG-14–321833 to QC]; and the Charles and Johanna Busch Biomedical Award [to QC].

## Additional information

### Funding

| Funder | Grant reference number | Author |
| --- | --- | --- |
| National Institutes of Health | R00AG033658 | Qian Cai |
| Alzheimer's Association | NIRG-14-321833 | Qian Cai |
| Rutgers University | the Charles and Johanna | Qian Cai |

|  | Busch Biomedical Award |  |
| --- | --- | --- |
| National Institutes of Health | R01NS089737 | Qian Cai |

The funders had no role in study design, data collection and interpretation, or the decision to submit the work for publication.

## Author contributions

PT, Conceptualization, Data curation, Formal analysis, Validation, Investigation, Visualization, Methodology, Project administration; XY, Data curation, Formal analysis, Validation, Investigation, Visualization, Methodology, Project administration; TF, Data curation, Formal analysis, Validation, Visualization, Methodology; DA, Data curation, Investigation, Visualization; QC, Conceptualization, Resources, Data curation, Software, Formal analysis, Supervision, Funding acquisition, Validation, Investigation, Methodology, Writing—original draft, Project administration, Writing—review and editing

## Author ORCIDs

Qian Cai, http://orcid.org/0000-0001-8525-2749

## Ethics

Animal experimentation: The animal care and use in this study was in accordance with Rutgers University IACUC standards and the protocol (#11-026) was approved by the Rutgers Animal Care and Use Committee.

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
