## [Decision Letter]

Thank you for submitting your article "Impaired retrograde transport of axonal autophagosomes contributes to autophagic stress in Alzheimer's disease neurons" for consideration by *eLife*. Your article has been favorably evaluated by Anna Akhmanova (Senior Editor) and two reviewers, one of whom, Hong Zhang, is a member of our Board of Reviewing Editors.

The reviewers have discussed the reviews with one another and the Reviewing Editor has drafted this decision to help you prepare a revised submission.

Summary:

This paper investigated the mechanism by which autophagic vacuoles accumulate in axons of Alzheimer's disease (AD). The authors showed that intracellular Abeta interferes with dynein and snapin/autophagosome interactions, thus stalling the vesicles. Biochemical analysis suggests a decrease of dynein intermediate chains (DIC) on immunoisolated late endosomes from AD transgenic mice. The human data also support the idea that autophagosomes accumulate in AD, and perhaps also that vesicle-associated snapin is reduced. The authors further demonstrated that overexpressing Snapin in hAPP neurons enhances AV retrograde transport and concomitantly reduces autophagic accumulation at presynaptic terminals. This study reveals a novel mechanism underlying the impairment of autophagy by pathogenic Abeta. Overall, the work is interesting. The reviewers found that additional work is required to strengthen the manuscript to meet the standards for *eLife*.

Essential revisions:

1) The Abeta used in the in-vitro experiments should be the same Abeta that is detected by the A11 antibody around the plaques. In Figure 3, the authors use A11, which is an antibody that sees soluble oligomeric (and not monomeric or fibrillar) Abeta; yet they refer to this as "soluble Abeta" (not soluble oligomer, or soluble monomer). Later for in-vitro biochemistry (Figure 4), they use synthetic Abeta 1-42, which is not representative of what was seen by the A11 antibody (though for the experiments to make sense, this needs to be the case). If possible, the same Abeta should be used for in-vitro experiments. At the least, the paper needs be written differently to highlight this caveat. Also, the reviewer found that the term "soluble Abeta1-42" is confusing. It forms aggregates that colocalize with p62 and ubiquitin. Please describe the properties of Abeta1-42 in more detail.

2) Autophagosome maturation in mammalian cells involves fusion of autophagosomes with early endosomes and late endosomes (LEs). Here the authors characterized the co-labeling of LC3 with p62 and LEs. Colocalization of LC3 with early endosomes in AD mouse brains or cultured neurons from mutant hAPP Tg mice could also be examined.

---

## [Author Response]

*Essential revisions:*

*1) The Abeta used in the in-vitro experiments should be the same Abeta that is detected by the A11 antibody around the plaques. In Figure 3, the authors use A11, which is an antibody that sees soluble oligomeric (and not monomeric or fibrillar) Abeta; yet they refer to this as "soluble Abeta" (not soluble oligomer, or soluble monomer). Later for in-vitro biochemistry (Figure 4), they use synthetic Abeta 1-42, which is not representative of what was seen by the A11 antibody (though for the experiments to make sense, this needs to be the case). If possible, the same Abeta should be used for in-vitro experiments. At the least, the paper needs be written differently to highlight this caveat. Also, the reviewer found that the term "soluble Abeta1-42" is confusing. It forms aggregates that colocalize with p62 and ubiquitin. Please describe the properties of Abeta1-42 in more detail.*

We apologize for not making it clearer with regard to the term of the “soluble Aβ1-42”. For our in vitro biochemical experiments, synthetic Aβ1-42 was prepared as soluble Aβ1-42 oligomers, the toxic form of Aβ linked to AD pathogenesis. Our new data showed that dynein intermediate chain (DIC) specifically interacts with this toxic form of Aβ1-42, but not Aβ1-40 (Figure 4).

We further demonstrated that oligomeric Aβ1-42 impairs recruitment of dynein motors to Snapin-associated LEs/amphisomes by competitively interrupting dynein DIC-Snapin interaction (Figure 4).

As the reviewers correctly noted, A11 antibody detects soluble Aβ oligomers, but not soluble monomer or insoluble fibrils (Jimenez et al., 2008; Jimenez et al., 2011; Kayed et al., 2003; Zempel et al., 2010). Alternatively, we utilized the well-characterized anti-β-amyloid 1-42 antibody (AB5078P), which only recognizes soluble Aβ1-42 oligomers, but not Aβ1-40 or high molecular weight insoluble forms of Aβ1-42 (Agholme et al., 2012; Kamal et al., 2001; Muresan et al., 2009; Takahashi et al., 2013). The observations using AB5078P antibody are consistent with those detected by A11 antibody (Figure 3—figure supplement 3). Taken together, our findings are representative of what was seen by both A11 and AB5078P antibodies, which allow us to propose that soluble Aβ1-42 oligomers interfere with dynein-Snapin motor-adaptor coupling, thereby impairing retrograde transport of autophagic vacuoles (AVs) and inducing autophagic stress in distal AD axons.

To examine the properties of Aβ1-42, we applied both A11 and AB5078P antibodies that both recognize soluble forms of oligomeric Aβ, but not insoluble aggregates or extracellular plaques. With these antibodies, we further showed that soluble Aβ1-42 oligomers associate with amphisomes and are enriched within distal axons and presynaptic terminals, but are not present in insoluble fibrils associated with amyloid plaques (Figure 3—figure supplement 3).

Consistently, AVs were associated with or surrounzded by oligomeric Aβ gold particles in the axon of hAPP neurons (Figure 3). Moreover, a significant number of LC3-marked AVs were co-labeled with p62 or Ubiquitin in the hippocampal mossy fibers and within swollen/dystrophic axons surrounding amyloid plaques (p62: 96.29% ± 0.43%; Ubiquitin: 97.89% ± 0.23%) (Figure 1—figure supplement 1). Thus, the co-localization with p62 and Ubiquitin most likely represents the association of soluble Aβ1-42 oligomers with p62 or Ubiquitin-labeled AVs, but not insoluble aggregates.

At the reviewers’ request, we described the properties of soluble Aβ1-42 in more detail and clarified these Aβ terms in the subsection “Association of Soluble Aβ Oligomers with Amphisomes in the Dystrophic Axons of AD Mice”.

*2) Autophagosome maturation in mammalian cells involves fusion of autophagosomes with early endosomes and late endosomes (LEs). Here the authors characterized the co-labeling of LC3 with p62 and LEs. Colocalization of LC3 with early endosomes in AD mouse brains or cultured neurons from mutant hAPP Tg mice could also be examined.*

The reviewers suggested a great experiment. In the revision, we examined the co-localization of LC3 with Rab5-labeled early endosomes in cultured neurons from mutant hAPP Tg mice. We found that about 46% of LC3-labeled AVs co-localized with early endosomes within the axon of mutant hAPP neurons (45.58% ± 2.24%; n=47, v=875). However, Rab5-marked early endosomes moved either a short distance, or in an oscillatory pattern along axons. While our observation is consistent with the results from previous studies (Cai et al., Neuron 2010; Chen and Sheng, J Cell Biol, 2013), the motility of axonal early endosomes showed no significant change in mutant hAPP neurons relative to that of WT neurons (WT: 67.53% ± 1.97; hAPP: 70.93% ± 2.31%; p=0.268). Accumulating evidence suggests that Rab5-endosomes usually mature into late endosomes (LEs) through acquisition of Rab7 and loss of Rab5 (Lakadamyali et al., Cell, 2006; Huotari and Helenius, EMBO J, 2011). Moreover, a recent study reported that nascent AVs gain retrograde transport motility by recruiting LE-loaded dynein-Snapin motor-adaptor complexes after fusion with Rab7-associated LEs to form amphisomes (Cheng et al., J Cell Biol, 2015). Thus, our data supports the notion that fusion of AVs with Rab5- endosomes could be a transitional process before they further mature into Rab7-positive amphisomes to gain long-distance retrograde transport motility. Consistently, our current study demonstrated that Aβ-mediated interruption of dynein-Snapin coupling leads to Rab7- amphisome stalling in the axon of AD neurons. Elevated Snapin expression rescues defects in retrograde transport of Rab7-amphisomes, and thus reduces autophagic stress in AD axons. We added these quantitative co-localization and motility data of early endosomes and AVs as revised Figure 2—figure supplement 2H-Jand described the results in the subsection “Impaired Retrograde Transport of Axonal Amphisomes in mutant hAPP Tg Neurons”, last paragraph and in the figure legend.